# The Value of Prediction in Identifying the Worst-Off

Unai Fischer-Abaigar [1 2]   Christoph Kern [1 2]   Juan Carlos Perdomo [3]

## Abstract

Machine learning is increasingly used in government programs to identify and support the most vulnerable individuals, prioritizing assistance for those at greatest risk over optimizing aggregate outcomes. This paper examines the welfare impacts of prediction in equity-driven contexts, and how they compare to other policy levers, such as expanding bureaucratic capacity. Through mathematical models and a real-world case study on long-term unemployment amongst German residents, we develop a comprehensive understanding of the relative effectiveness of prediction in surfacing the worst-off. Our findings provide clear analytical frameworks and practical, data-driven tools that empower policymakers to make principled decisions when designing these systems.

## 1. Introduction

Faced with pressure to modernize, large bureaucracies are increasingly adopting risk prediction tools to improve efficiency and better serve their populations. Beyond optimizing aggregate outcomes, investments in these programs often aim to address historical inequities and prioritize the needs of the worst-off. For instance, in 2012, Wisconsin launched a risk prediction system to explicitly address deep racial disparities in academic achievement and improve high school graduation rates amongst underserved students. More broadly, such systems are particularly relevant in settings where normative considerations demand prioritizing those at the greatest risk of adverse outcomes, and where well-established downstream interventions can meaningfully benefit these vulnerable individuals.

From a design perspective, these risk predictors are challenging to evaluate because their value cannot be assessed without reference to the broader social context. The value of a risk predictor is ultimately determined by its impact on bottom-line welfare (e.g., graduation rates) and how these welfare impacts compare to those of other bureaucratic alternatives (Johnson & Zhang, 2022). For example, to understand whether investments in prediction are truly valuable in Wisconsin, we need to assess how much better the risk predictor is at identifying at-risk students relative to existing policies. We also need to understand whether sophisticated prediction systems yield higher graduation rates amongst the underserved than structural investments in teacher training or better facilities.

Equity-driven programs are pervasive in applications like social housing, poverty targeting, and unemployment assistance. In these contexts, many government agencies are exploring how algorithmic prediction systems may be an improvement over their current profiling processes (Körtner & Bonoli, 2023). Yet, due to the absence of an overarching framework that allows the systematic assessment of the relative impacts of different design decisions, efforts to improve predictive accuracy are rarely studied in concert with other policy levers such as expanding screening capacity.

Building on recent work in a budding area of learning in resource allocation contexts, we develop tools to evaluate the design and broader impact of prediction systems that aim to identify the worst-off members of a population. We develop a holistic understanding of the value of statistical prediction in these contexts through theoretical insights into foundational statistical models and a real-world case study on identifying long-term unemployment. Our results establish clear theoretical and empirical criteria characterizing the relative value of core design decisions within these problems. Specifically, we identify when improving prediction provides a higher marginal benefit in helping an institution identify the worst-off. This is compared to alternative strategies, such as keeping prediction accuracy fixed, expanding bureaucratic capacity and screening a larger population.

Interestingly, we show that prediction is a first and last-mile effort. The impacts of improving prediction are always outweighed by those of expanded screening capacity, except for when the system explains either none or almost all of the variance in outcomes. While this relationship is moderated by costs, it still largely holds when prediction improvements

[1]Department of Statistics, University of Munich (LMU), Munich, Germany [2]Munich Center for Machine Learning, Germany [3]Harvard University, Boston, MA, US. Correspondence to: Unai Fischer-Abaigar <Unai.FischerAbaigar@stat.uni-muenchen.de>.

*Proceedings of the 42nd International Conference on Machine Learning*, Vancouver, Canada. PMLR 267, 2025. Copyright 2025 by the author(s).

are more cost-efficient than measures that expand access.

These results are counternarrative to current efforts in empirical public policy where agencies focus on incremental improvements within complex prediction systems, starting from the solid baseline performances of their current processes (Desiere et al., 2019; Desiere & Struyven, 2021). Furthermore, implementing more complex profiling systems at scale comes with operational costs (such as staff training and data collection) which need to be contextualized by the cost-benefit ratio of expanding access. Our empirical case study explicates how to systematically assess the relative gains of these design components in a real-world application setting, translating formal insights into critical guidance for designers of these systems.

Our results provide theoretically principled and empirically grounded tools for policymakers to make informed decisions when designing prediction systems to identify the worst-off. They also offer a practical framework to help determine how much should be invested in prediction relative to other interventions and how to decide when prediction systems are "good enough" for deployment.

### 1.1. Overview of Framework and Contributions

**Setup.** We consider a scenario where a decision-maker seeks to identify worst-off members of a population, as determined by a real-valued welfare metric $Y \in \mathbb{R}$, with the goal of prioritizing them for further screening and support. The population is represented by a distribution $\mathcal{D}$ over features $X$ and outcomes $Y$. The planner aims to identify all individuals whose outcomes $Y$ fall below some threshold $t(\beta)$, $Y \leqslant t(\beta)$. Here, $\beta \in [0, 1]$ is a parameter (quantile) that determines the size of the population that is at risk, $\Pr[Y \leqslant t(\beta)] = \beta$. For instance, in poverty prediction, $Y$ is income, and the goal is to identify everyone whose income is below some value.

To solve this problem, the social planner has access to data $(X, Y) \sim \mathcal{D}$ and builds a screening policy $\pi : \mathcal{X} \to \{0, 1\}$ that determines whether an individual with features $x$ is screened from the broader population to see if they belong to the worst-off group. Learning plays a fundamental role since the optimal policy is to predict each person's expected outcome, $f(x) = \hat{Y} \approx \mathbb{E}[Y \mid X = x]$ and screen those in the bottom fraction, $\pi_f(x) = 1\{f(x) \leqslant t(\alpha)\}$.

Unpacking this further, $\alpha \in [0, 1]$, is a design parameter that determines how many people the planner can screen, $\Pr[f(x) \leqslant t(\alpha)] = \alpha$. The amount of resources $\alpha$ need not be equal to the size of the target population $\beta$. For instance, an organization might have normative goal of identifying the poorest 5% of individuals, but only have the resource to screen 1% of the population. Conversely, they might realize that predictions are not perfect, and that to identify the bot-

tom 5%, they might have to screen 10% of the population.

Given a predictor $f$, a screening budget of $\alpha$, and a target parameter $\beta$, the value of a prediction system is equal to the fraction of the at-risk population that it identifies,

$$V(\alpha, f; \beta) = \Pr_{\mathcal{D}}[f(x) \leqslant t(\alpha) \mid Y \leqslant t(\beta)],$$

where again $t(\alpha), t(\beta)$ are chosen to respect the design constraints. We focus on this notion of value since our driving motivation is to analyze domains like unemployment assistance, or poverty prediction, where there is no harm in the prediction system raising a false positive ($\pi(x) = 1, Y > t(\beta)$). By and large, the true value of the system is equal to the extent that it helps an institution efficiently identify the needy amongst a large, diverse population.

The focus of our work is to build a holistic understanding of prediction in these contexts by evaluating the relative impacts of different design parameter, such as expanding screening capacity or improving prediction, on this notion of bottom-line value $V(\alpha, f; \beta)$. We develop these insights through theoretical investigations as well as in-depth empirical case study.

**Mathematical Results.** Following Perdomo (2024), we formalize the relative value of prediction for the worst-off by studying the *prediction-access ratio* or PAR. Intuitively, the PAR measures the relative change in value achieved by optimizing different policy levers.

$$\text{PAR} = \frac{\text{Marginal Value of Expanding Access}}{\text{Marginal Value of Better Prediction}}.$$

We formally define this quantity in Equation 3. While initially developed to specifically study the value of prediction in allocation problems where allocating goods to individuals had heterogeneous effects, here we extend this concept to analyze the value of prediction in a related, but distinct, setting where we aim to identify the worst-off.

Small values of the PAR (i.e. PAR $< 1$) indicate that small improvements in prediction yield a much larger (relative) impact in the ability to target the worst-off than a small expansion in screening capacity. The opposite is true if the PAR is greater than 1. Calculating this quantity is a fundamental step in deciding which policy lever makes economic sense.

**Costs.** A full cost-benefit analysis requires combining the prediction-access ratio with the (marginal) costs of improvements in capacity $C_{\text{Access}}$ and prediction $C_{\text{Pred}}$. Once we factor in costs, it is easy to decide what to focus on. A social planner should expand access whenever

$$\frac{C_{\text{Access}}}{C_{\text{Pred}}} < \text{PAR}$$

and invest in better prediction otherwise. The (marginal) costs that competing policy levers carry are inherently

context-dependent and will vary across application domains. In many applied settings the cost ratio is comparatively well understood, for example, the salary of an additional case-worker or the cost of a household survey. By presenting PAR separately from costs, we isolate the welfare side of the equation; domain experts can then plug in their own cost estimates to reach a policy decision. In particular, the PAR tells us how much we should be willing to *pay* for improvements in prediction versus expanding access.

We encourage future work to explore scenarios with more complex or less clearly defined cost structures. For instance, many practical applications involve recurring costs, such as ongoing staff salaries or periodic data collection, and fixed costs, such as initial investments in infrastructure or predictive model development. Analyzing how these cost structures affect welfare decisions over time, including amortization of fixed investments or identifying the point at which specific improvements become cost-effective, would significantly enhance our understanding of the relative value of prediction.

To build intuition for the value of prediction in identifying the worst-off, we examine the prediction access ratio in one of the most basic statistical models. The outcomes $Y$ are Gaussian, and the learner has access to a predictor $f(x) = \hat{Y}$ such that the errors $Y - \hat{Y}$ are also Gaussian and independent of $\hat{Y}$. While extremely simple, the model yields surprisingly precise numerical insights that exactly match up in our real-world case study, where, of course, none of these assumptions hold. In this setting the quality of $\hat{Y}$ is fully summarized by the coefficient of determination $R^2 = \text{corr}(Y, \hat{Y})^2$.

Our first result identifies when local improvements in prediction have the highest impact:

**Theorem 1.1** (Informal, see Theorem 3.2). *If $\alpha$ is at least a constant, the local improvements in $V$ with respect to $R^2$ diverge in two regimes: (1) $R^2 \to 1$ and $\alpha = \beta$, or (2) $R^2 \to 0$. In both cases, the prediction-access ratio satisfies* $\text{PAR}(\alpha, \beta) = 0$.

Predictions have the highest marginal impact at low and high $R^2$-values, making them a first- and last-mile effort. Our second result characterizes when the opposite is true. We prove that whenever screening capacities are severely limited relative to the size of the population one aims to identify $\alpha \ll \beta$, the benefits of increasing $\alpha$ are overwhelming. Furthermore, it shows that the impacts of improving access are still relatively larger exactly in the regime where most current systems operate: $f$ explains $\approx 20\%$ of the variance and $\alpha$ is equal to, or even slightly larger, than $\beta$.

**Theorem 1.2** (Informal, see Theorem 3.1, Proposition 3.3). *If the predictor $f$ explains an $R^2$ fraction of the variance, where $R^2$ is at least a constant, then the prediction access ratio is at least $\Omega(\alpha^{-1/(1-R^2)})$. Furthermore, if $0.15 \leqslant R^2 \leqslant 0.85$ and $\alpha \leqslant \beta$ or $0.2 \leqslant R^2 \leqslant 0.5$, $\beta \geqslant 0.15$, and $\alpha \leqslant 0.5$ then the local prediction-access ratio is at least 1.*

**Empirical Results.** We complement our theoretical discussion by presenting a methodology for policymakers to evaluate the prediction-access ratio in practice. Using a real-world administrative dataset on hundreds of thousands of jobseekers in Germany, we show that our theoretical findings generalize to a more complex, real-world context that closely resembles algorithmic profiling systems widely implemented in many countries. Notably, our results reveal that when considering non-local improvements, expanding screening capacity has an even greater impact compared to enhancing prediction accuracy.

## 1.2. Related Work

Machine learning is increasingly used in the public sector to allocate support by predicting individuals at risk of adverse outcomes (Fischer-Abaigar et al., 2024), with applications spanning a wide range of problem domains (Desiere et al., 2019; Blumenstock, 2016; Perdomo et al., 2023; Chan et al., 2012; Potash et al., 2015; Chouldechova et al., 2018). A large methodological literature draws on decision theory, operations research, economics, and machine learning to learn allocation rules from data (Elmachtoub & Grigas, 2022; Kitagawa & Tetenov, 2018; Manski, 2004; Fernández-Loría & Provost, 2022), with recent work in causal inference focusing on learning treatment policies from observational data (Athey & Wager, 2021; Kallus, 2021). However, many decision-makers rely on separately trained predictive risk scoring-systems to solve "prediction policy problems" (Kleinberg et al., 2015). Recently, this work has been extended using causal inference to train and evaluate these systems (Coston et al., 2023; Guerdan et al., 2023; Boehmer et al., 2024).

The widespread use of risk-scoring systems has raised concerns regarding their tradeoffs, pitfalls, and validity (Wang et al., 2024; Coston et al., 2023; Fischer-Abaigar et al., 2024). These concerns include not only questions of empirical performance but also of fairness and equity in how predictive systems shape access to public services (Barocas et al., 2023). Recent work explores alternative design choices—such as employing aggregate rather than individual-level predictions (Shirali et al., 2024), balancing immediate needs with information-gathering (Wilder & Welle, 2024), and introducing randomization (Jain et al., 2024)—to improve downstream outcomes.

Perdomo (2024) studies the prediction-access ratio under both linear and probit models, with the latter closely related to our work. While they focus on binary welfare outcomes, we adopt a continuous welfare metric and a distinct policy objective: rather than evaluating changes in overall expected

welfare, we measure the fraction of truly worst-off individuals who are identified. This captures a mathematically and conceptually distinct setting frequently encountered in the public sector. For instance, employment agencies often prioritize identifying and assisting individuals in greatest need, rather than optimizing average employment outcomes across all jobseekers. In addition, we introduce a set of empirical tools to analyze these tradeoffs in practice, while the work of Perdomo (2024) is purely theoretical.

## 2. Formal Framework

We start by formally defining our screening problem.

**Definition 2.1** (Screening Problem). The screening problem seeks to identify a decision rule $\pi\colon \mathbb{R} \to \{0,1\}$ that fraction of the worst-off population that is identified while adhering to resource constraints $\alpha \in (0,1)$ that bound the percentage of the population that can be screened by the social planner:

$$\max_{\pi\colon \mathbb{R}\to\{0,1\}} \mathbb{E}\left[\pi(\hat{Y}) = 1 \mid Y \leqslant F_Y^{-1}(\beta)\right] \text{ s.t. } \mathbb{E}\left[\pi(\hat{Y})\right] \leqslant \alpha$$

The quantile $F_Y^{-1}(\beta)$ denotes the welfare cutoff that identifies the worst-off $\beta \in (0,1)$ fraction of the population.

Given perfect knowledge of the welfare outcomes $\hat{Y} = Y$, the optimal decision policy is simple: rank individuals based on their outcomes $Y$ and intervene in the bottom $\alpha$-fraction of the population. In the general case, we have:

**Proposition 2.2.** *The optimal policy $\pi^*\colon \mathbb{R} \to \{0,1\}$ to solve the screening problem (Definition 2.1) is equal to $\pi^*(\hat{Y}_i) = 1\{s(\hat{Y}_i) \geqslant F_s^{-1}(1-\alpha)\}$ where $F_s^{-1}(1-\alpha)$ is the $(1-\alpha)$-quantile of $s(\hat{Y}) = \Pr[Y \leqslant F_Y^{-1}(\beta) \mid \hat{Y}]$.*

**Policy Value in Gaussian Setting.** For the theoretical investigation, we assume independent, identically distributed errors $\varepsilon = Y - \hat{Y} \overset{iid}{\sim} \mathcal{N}(0,\gamma^2)$ that are independent of $\hat{Y}$. In this setting, the screening problem can be solved by ranking individuals in ascending order of their predicted outcomes $\hat{Y}$ and screening the bottom $\alpha$-fraction (see Proposition C.1), achieving the policy value:

$$V(\pi^*) = \Pr[\hat{Y} \leqslant F_{\hat{Y}}^{-1}(\alpha) \mid Y \leqslant F_Y^{-1}(\beta)] \quad (1)$$

In addition, we assume welfare outcomes $Y \sim \mathcal{N}(\mu,\eta^2)$. Because $\varepsilon$ is independent of $\hat{Y}$, this implies that $Y$ and $\hat{Y}$ follow a bivariate normal distribution.

**Proposition 2.3.** *(Policy Value in Gaussian Setting) Let $Y - \hat{Y} \overset{iid}{\sim} \mathcal{N}(0,\gamma^2)$ and $Y \sim \mathcal{N}(\mu,\eta^2)$, then the value $V(\pi^*)$ of the optimal screening policy $\pi^*$ is given by*

$$V(\pi^*) = V(\alpha,\beta,R^2) = \frac{\Phi_2\left(\Phi^{-1}(\alpha),\Phi^{-1}(\beta);\rho\right)}{\beta} \quad (2)$$

*where $\Phi_2(\cdot)$ denotes the bivariate standard normal CDF with correlation $\rho = \sqrt{\eta^2 - \gamma^2}/\eta$ and $\Phi^{-1}(\cdot)$ is the quantile function of the standard normal distribution.*

In this model, the goodness of the predictions $\hat{Y}$ are entirely captured by the coefficient of determination $R^2$, which equals the squared correlation $\rho^2$ between $Y$ and $\hat{Y}$.

Our analysis extends to the log-normal distribution $\log Y \sim \mathcal{N}(\mu,\eta^2)$ under a a multiplicative error model $Y = \hat{Y} \cdot u$ with $\log u \sim \mathcal{N}(0,\gamma^2)$. Taking logarithms, leads to $\log Y = \log \hat{Y} + \log u$. Since the logarithm is strictly increasing, the ordering of $Y$ and $\hat{Y}$ is preserved under transformation. This allows us to apply the same framework to the log-transformed variables $\log Y$ and $\log \hat{Y}$. This extension is particularly useful because many welfare outcomes, such as income distributions (Clementi & Gallegati, 2005), can be approximated by a log-normal distribution.

**Visualization.** For a given screening capacity $\alpha$ and $R^2$ value, we can illustrate the corresponding screening policy by plotting the probability $\Pr\{\hat{Y} \leqslant F_{\hat{Y}}^{-1}(\alpha) \mid Y = y\}$ that an individual with welfare outcome $Y = y$ is screened. As shown in Figure 1, lower values of $Y$ correspond to higher probabilities of being screened. We focus on evaluating how effectively the screening policy identifies individuals in the worst-off segment of the population (i.e., on the left side of the $\beta$ cutoff).

## 3. Theoretical Results

The decision-maker has (at least) two pathways to raise the policy value, which we refer to as *policy levers*:

- **Expanding Access** Increasing the screening threshold from $\alpha$ to $\alpha + \Delta_\alpha$. If full screening were possible ($\alpha = 1$), the $\beta$-fraction would be fully identified, as $V(\pi^*) = \frac{\Phi_2\left(\Phi^{-1}(\alpha),\Phi^{-1}(\beta);\rho\right)}{\beta} = \frac{\Phi\left(\Phi^{-1}(\beta)\right)}{\beta} = 1$.
- **Improving Predictions** Investing in better predictive models, modeled as increasing $R^2$ to $R^2 + \Delta_{R^2}$. Perfect predictions ($R^2 = 1$) leads to optimal allocation of available capacities: $V(\pi^*) = \frac{1}{\beta}\Phi\left(\min(\Phi^{-1}(\alpha),\Phi^{-1}(\beta))\right)$.

Figure 1 showcases improvements in access and prediction. Increasing capacity expands the fraction of the population screened, while improving $R^2$ shifts probability mass across the $\beta$ threshold, enhancing targeting accuracy.

Following Perdomo (2024), a key quantity of interest is the *prediction-access ratio* (PAR), which quantifies the relative improvements in policy value from enhancing predictions versus improving access to screening. Specifically, the PAR is defined as:

$$\text{PAR} = \frac{V(\alpha+\Delta_\alpha,\beta,R^2)-V(\alpha,\beta,R^2)}{V(\alpha,\beta,R^2+\Delta_{R^2})-V(\alpha,\beta,R^2)} \quad (3)$$

In other words, the PAR can inform a social planner how much more they should be willing to pay for improvements in screening capacity relative to prediction. For example, a PAR $> 2$ implies that expanding the screening capacity by

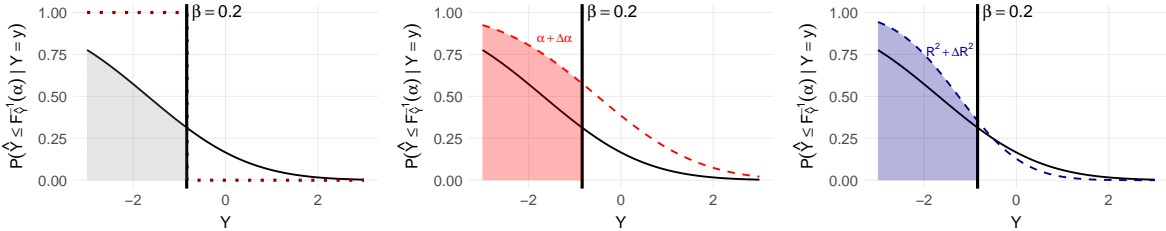

*Figure 1.* **Screening Policy in Gaussian Setting.** (Left) Probability of being screened for an individual with a specific welfare outcome $Y$, given $R^2 = 0.25$, $\alpha = 0.2$, and $\beta = 0.2$. The dashed line represents the unconstrained oracle policy, which perfectly screens those in need. (Middle) Policy with expanded screening capacity, where $\alpha$ increases by $\Delta_\alpha = 0.2$. (Right) Policy under an improved prediction model with $R^2 + \Delta_{R^2}$, where $\Delta_{R^2} = 0.2$. The shaded area under $\Pr[\hat{Y} \leqslant F_{\hat{Y}}^{-1}(\alpha) \mid Y = y]$, weighted by $f_Y(y)$ and normalized by $\Pr[Y \leqslant F_Y^{-1}(\beta)]$, corresponds to the policy value.

$\Delta_\alpha$ yields at least twice the increase in policy value compared to investing in improved predictions by $\Delta_{R^2}$. Consequently, the social planner should prioritize investments in screening capacity, provided the costs of doing so are not more than double those of improving predictions.

### 3.1. Theoretical Bounds for the Prediction-Access Ratio

In our setting, direct calculation of the PAR is challenging due to the policy value being analytically intractable and the problem featuring strong non-linearities. We derive bounds for specific cases and regimes that we consider particularly insightful, with a focus on marginal local improvements. In our empirical investigation, we find that the main results generalize well to a more complex, real-world setting.

**What should priorities be if screening is very limited?**

**Theorem 3.1** (PAR for Small Screening Capacities)**.** *For any $0 < R^2 < 1$, $\Delta_{R^2}, \Delta_\alpha > 0$ and $0 < \beta \leqslant 0.5$ there exists a threshold $t(\beta, R^2, \Delta_{R^2})$ such that for any $\alpha + \Delta_\alpha \leqslant t$, $\mathrm{PAR}(\alpha, R^2, \Delta_\alpha, \Delta_{R^2})$ is at least*

$$\frac{\Delta_\alpha}{\Delta_{R^2}} \sqrt{R^2(1 - R^2)} \left(5.1 \cdot \alpha \Phi^{-1}(1 - \alpha)\right)^{-\frac{1}{1 - R^2} + o(1)}$$

*where $o(1)$ goes to zero as $\alpha$ approaches zero.*

Suppose the available screening capacity $\alpha + \Delta_\alpha$ is very small ($\alpha + \Delta_\alpha \ll \beta$), and assume there is a baseline level of predictability (i.e., $R^2$ is bounded away from 0). Then Theorem 3.1 implies that the PAR can become very large. Specifically, for small $\alpha$, $\Phi^{-1}(1 - \alpha)$ grows asymptotically like $\sqrt{\log(1/\alpha)}$. Consequently, the polynomial growth of $\alpha^{-1/(1-R^2)}$ drives the PAR to increase rapidly as $\alpha$ decreases. It follows that in the scarce capacity regime, expanding the screening capacity has a far greater impact than improvements in prediction accuracy.

**When does prediction have the highest impact?**

**Theorem 3.2** (Maximally Effective (Local) Prediction Improvements)**.** *Let $0 < \beta < 1$ be fixed and $0 < \alpha < 1$. Consider the points that maximize the local rate of change*

*in policy value $V$ with respect to improvements in $R^2$:*

$$(\alpha_*, R_*^2) = \underset{(\alpha, R^2) \in (0,1) \times (0,1)}{\arg\max} \lim_{\Delta \to 0} \frac{V(\alpha, \beta, R^2 + \Delta) - V(\alpha, \beta, R^2)}{\Delta}$$

*The local improvements in $V$ diverge—and are maximized—in two regimes: (1) $R_*^2 \to 1$, $\alpha_* = \beta$, and (2) $R_*^2 \to 0$. For both regimes, setting $\Delta_{R^2} = \Delta_\alpha = \Delta$, the local prediction-access ratio satisfies $\lim_{\Delta \to 0} \mathrm{PAR}(\alpha, \beta, \Delta) \to 0$.*

According to Theorem 3.2, marginal improvements in prediction are most impactful in two distinct regimes. First, when predictive capacity is very low, even a small initial investment can lead to disproportionately large improvements, provided that a minimal baseline of screening capacity is present. Second, as $R^2$ approaches one, further marginal improvements can also have a significant relative impact, specifically around the point where the screening capacity $\alpha$ matches the requirements for screening the entire $\beta$-segment of the population. See Figure 2.

**When are small increases in screening capacity more impactful than improving predictions?**

**Proposition 3.3** (PAR for Local Improvements)**.** *Let $R^2$, $\beta$, and $\alpha$ satisfy either $R^2 \in (0.15, 0.85)$, $\beta \in (0.03, 0.5)$, and $\alpha \leqslant \beta$, or $R^2 \in (0.2, 0.5)$, $\beta \geqslant 0.15$, and $\alpha \leqslant 0.5$. If $\Delta_{R^2} = \Delta_\alpha = \Delta$, then $\lim_{\Delta \to 0} \mathrm{PAR}(\alpha, \beta, \Delta) \geqslant 1$.*

We find that the PAR remains above one as long as $\alpha \leqslant \beta$ and $R^2$ is not too extreme. For larger $\beta$ values (i.e., $\beta \geqslant 0.15$) the PAR stays above one even for large $\alpha$ provided $R^2$ remains in a moderate range. Crucially, this represents the standard parameter regime in which most allocation programs operate, characterized by a moderate baseline of predictions and resource levels comparable to $\beta$.

**Numerical Simulations.** We complement our theoretical investigation with numerical simulations of the PAR for different $\alpha$, $\beta$ and $R^2$ values (see Figure 2). Consistent with our theoretical results, the PAR becomes large for small screening capacities ($\alpha \ll \beta$) and remains above one for $\alpha \leqslant \beta$, provided a small baseline level of predictive performance has been established. The bounds in Proposition 3.3

are conservative, with PAR $> 1$ observed for a broad range of $R^2$ values. Prediction improvements are particularly impactful when $R^2$ is small. Although the PAR falls below one in the high-$R^2$ and high-$\alpha$ regime, allocation is nearly perfect, making further improvements a "last mile" effort.

**Discussion.** We found several insights relevant to policymakers aiming to iteratively improve a screening system. First, establishing a baseline level of predictive performance is usually a good starting point. Once this is achieved, expanding the screening capacity becomes the next priority. For very small capacities, Theorem 3.1 tell us that the PAR can increase significantly, making investments in screening capacity highly impactful.

Generally, expanding capacity to at least the level where everyone in need could hypothetically be screened ($\alpha \geqslant \beta$) is likely cost-efficient. Once both screening capacity and predictive accuracy are high and the allocation system is close to optimal, improvements in prediction become relatively more valuable again for perfecting the system. However, this regime may rarely be reached in practice.

In Figure 2, we display the PAR for a cost ratio of $1/4$. As expected, the regions where investing in $R^2$ is more efficient expand, and some of the earlier nonquantitative bounds no longer apply. Nevertheless, the key insights remain consistent: when screening capacities are small, investments in expanding them are very effective, while improvements in $R^2$ are more important when predictive accuracy is low.

## 4. Empirically Evaluating the PAR

While our theory offers broad intuition when expanding screening capacity or improving predictions is most effective, policymakers need practical tools for their own systems. To support this, we develop a methodology to compute and interpret the prediction-access ratio using empirical data, helping social planners identify the most efficient policy levers for their unique problem context.

**Policy Value.** As before, we define the allocation policy's value as the probability that the worst-off individuals are successfully identified:, i.e. $V(\alpha, \beta) = \Pr[\hat{Y} \leqslant F_{\hat{Y}}^{-1}(\alpha) \mid Y \leqslant F_Y^{-1}(\beta)]$. In practice, this can be measured using a recall-like metric, capturing the proportion of truly at-risk individuals screened by the policy.

$$V(\alpha, \beta) \approx \frac{\sum_{i=1}^n 1\{\hat{Y}_i \leqslant F_{\hat{Y},n}^{-1}(\alpha)\} 1\{Y_i \leqslant F_{Y,n}^{-1}(\beta)\}}{\sum_{i=1}^n 1\{Y_i \leqslant F_{Y,n}^{-1}(\beta)\}}$$

**Increasing Screening Capacity.** Given a chosen $\Delta_\alpha$ the policy improvement can be directly computed $V(\alpha + \Delta_\alpha, \beta) - V(\alpha, \beta)$ by recalculating the empirical policy value at the new threshold. For example, in cash transfer programs (Blumenstock, 2016), a key question is how many resources

$\alpha^*$ are required to reach a specified fraction $p$ of poor households, i.e. $\alpha^* = \inf_{\alpha \in (0,1)}\{\alpha \colon V(\alpha, \beta) \geqslant p\}$.

**Improving Predictions.** A decision-maker can improve a model's predictions through various pathways:

a) **Data Collection** Collect additional samples and increase the frequency of data collection. Social prediction systems are often vulnerable to distribution shifts over time in dynamic and evolving environments (Fischer-Abaigar et al., 2024; Aiken et al., 2023).

b) **Data Quality** Improve data quality (i.e., reduce errors and missing data) by means such as standardizing data collection processes, implementing centralized data management systems, and offering targeted training programs for staff.

c) **Collect Additional Features** In government, this may involve integrating separate data sources across institutions (Sun & Medaglia, 2019; Wirtz et al., 2019).

d) **Advanced Modeling Techniques** Utilize more sophisticated modeling techniques, which might capture more complex patterns in the data but are often more costly to operationalize.

In resource-constrained settings, planners often focus on incremental improvements rather than rebuilding entire systems. For instance, collecting a small amount of additional data may boost $R^2$ by a few points, uniformly reducing errors. To simulate such minor gains, we scale the model's residuals $\hat{Y}_+ = \hat{Y} + \delta(Y - \hat{Y})$, choosing $\delta \in (0, 1)$ so that $R^2$ increases by a target $\Delta_{R^2}$ (see Appendix B.3). This preserves the overall error structure, allowing us to gauge how a "similar but slightly better" model affects policy outcomes.

This approach can be extended in several ways. For example, residuals could be adjusted for specific subgroups to account for uneven prediction improvements (e.g., targeted data collection for rural or underrepresented populations). Alternatively, planners could retrain models under different conditions—such as sample size, feature set, or architecture—and compare the resulting policy value.

## 5. Case Study: Identifying Long-Term Unemployment in Germany

Public employment services (PES) across the globe make use of profiling approaches to identify jobseekers at risk of long-term unemployment to target preventative measures (Loxha & Morgandi, 2014). Starting from traditional rule-based approaches, many PES either test or already deploy algorithmic profiling to identify jobseekers in need of support (Desiere et al., 2019; Körtner & Bonoli, 2023). While these profiling tools assist in allocating programs that account for large shares of PES spending — making design choices critical (Kern et al., 2024) — systematic assessments of their relative value compared to other measures for

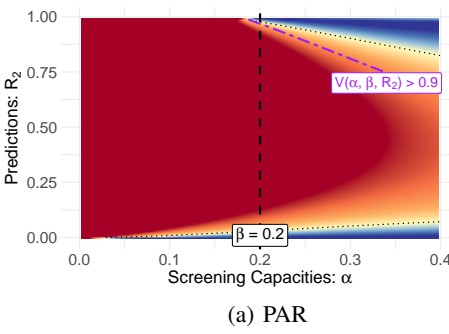
(a) PAR

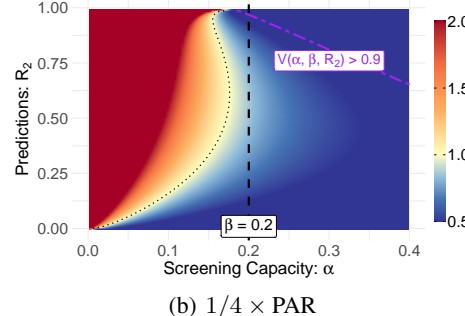
(b) $1/4 \times$ PAR

*Figure 2.* **Numerical Simulation of the Prediction-Access Ratio (PAR)**, Equation 3, for $\Delta_{R^2} = \Delta_\alpha = 0.01$ and $\beta = 0.2$. (Left) The PAR values. (Right) $1/4 \times$ PAR, representing a cost ratio of $1/4$. Each point represents a screening capacity $\alpha$ (x-axis) and $R^2$ value (y-axis), with the color bar showing the PAR clipped to the range $[0.5, 2.0]$. Dotted black lines represent PAR $= 1$, where improvements in $\alpha$ and $R^2$ are equally effective. The purple line marks the region in the $(\alpha, R^2)$ space where the policy value $V(\alpha, \beta, R^2)$ exceeds 0.9.

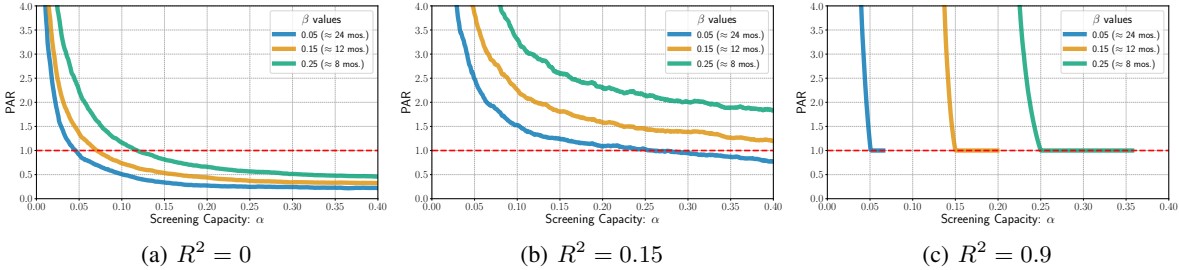
(a) $R^2 = 0$         (b) $R^2 = 0.15$         (c) $R^2 = 0.9$

*Figure 3.* **Prediction-Access Ratio** for $\Delta_{R^2} = \Delta_\alpha = 0.1$ across three regimes: (left) constant prediction ($R^2 = 0$), (middle) trained model ($R^2 = 0.15$), and (right) near-perfect prediction ($R^2 = 0.9$). As expected from our theoretical intuition, the PAR is large for small $\alpha$ and in the middle plot, which represents the typical regime for allocation systems.

improving jobseekers' outcomes remain absent.

We secured access to a dataset[1] on German jobseekers derived from German administrative labor market records that cover a large portion of the German labor force. It merges multiple administrative data sources, containing a wide spectrum of individual labor market information — including records on employment histories, received benefits, unemployment periods, participation in job training programs and demographic information. Such administrative records are the primary data source used by PES to build algorithmic profiling models (Bach et al., 2023).

**Experimental Setup.** We train a model to predict how long a newly registered jobseeker remains unemployed, defining the target $Y$ as unemployment duration in days (capped at 24 months). Following Bach et al. (2023), we use a set of covariates capturing demographic information, labor market history, and most recent job details. To ensure full 24-month observations and mimic a realistic deployment scenario, we focus on unemployment spells beginning between 2010 and

---

[1]The dataset is provided via a Scientific Use File by the Research Data Centre (FDZ) of the German Federal Employment Agency (BA) at the Institute for Employment Research (IAB) (Schmucker & vom Berge, 2023a;b).

2015, resulting in data on 274,515 different jobseekers and 553,980 unemployment spells. We refer to Appendix B.1 for additional information on the experimental setup and data.

Our focus is the $\beta$-fraction of jobseekers with the longest expected unemployment durations, representing those most at risk. In Germany, being unemployed for over one year (about $15\%$ of cases in our data; Figure 8) meets the legal definition of long-term unemployment (Bach et al., 2023), but some countries adopt different cutoffs (Desiere et al., 2019).

### 5.1. Results

We train a `CatBoost` model (see Appendix B.2 for details), achieving an $R^2$ of 0.15 on the test set. This level of predictive power aligns well with what is typically observed in social prediction tasks (Salganik et al., 2020) and similar applied settings (Desiere et al., 2019).

**How much does the screening capacity need to increase to target a significant fraction of high-risk jobseekers?** As expected, larger screening capacities increase both the policy value and the number of high-risk jobseekers screened (see Figure 4(a)). Focusing on the (German) LTU

cutoff ($\beta \approx 0.15$), our policy value aligns well with findings of previous studies[2] (Bach et al., 2023).

A planner might begin by setting $\alpha = \beta$, ensuring that, in theory, enough capacity is provided to screen and support every high-risk jobseeker. A natural question then arises: how much additional capacity $\Delta_\alpha$ would be required to screen at least a specified percentage of high-risk individuals? This additional capacity represents the overhead that must be invested to account for imperfect predictions. We observe that the $\Delta_\alpha$ required to ensure at least 75% of high-risk jobseekers are screened remains consistently around 0.25 across different $\beta$ values. While the policy value increases as $\alpha = \beta$ rises, the marginal improvements gained from increasing access decrease for higher $\alpha$, resulting in a somewhat stable $\Delta_\alpha$ across $\beta$. In practice, this means we need to screen 25% more of the population to ensure adequate coverage.

**What is the impact of improving screening capacity versus prediction errors?** We simulate small improvements in the $R^2$ value by uniformly scaling the residuals by a multiplicative factor. To ensure that this approach approximates a realistic pathway of (marginally) improving the model, we train various models at different sample sizes. We then verify that as $R^2$ increases with the amount of training data, the variance of the residuals decreases, while the distribution remains largely unchanged in shape (see Figure 12). We then evaluate the prediction-access ratio for $\Delta_{R^2} = \Delta_\alpha = 0.1$ in three scenarios : (1) the trained CatBoost model with $R^2 = 0.15$, (2) near-perfect predictions with $R^2 = 1 - \Delta_{R^2}$ and (3) constant predictions ($R^2 = 0$), effectively randomizing screening decisions.

We observe a rise in the PAR for small screening capacities $\alpha$ (see Figure 3), consistent with Theorem 3.1. Under random allocation ($R^2 = 0$), the PAR stays below one for $\alpha \geqslant 0.1$. This result aligns somewhat with Theorem 3.2, where we found that the (local) PAR approaches zero as $R^2 \to 0$. Because we consider $\Delta = 0.1$ (rather than an infinitesimal improvement, see Figure 13 for $\Delta = 0.01$), the PAR remains large at small $\alpha$. For the CatBoost model ($R^2 = 0.15$), capacity improvements stay relatively more effective (i.e., PAR > 1) for larger $\alpha$, matching Proposition 3.3, where we found that for moderate $R^2$ and $\alpha \leqslant \beta$, the local PAR remains above one. Meanwhile, near-perfect predictions ($R^2 = 0.9$) make capacity investments highly efficient, causing the PAR to diverge for $\alpha < \beta$, then drop sharply near $\alpha = \beta$ because the allocation becomes nearly optimal. When $\alpha \geqslant \beta$, the PAR stabilizes at one as numerator and denominator both approach zero.

These observations broadly match our theoretical findings,

despite the non-local improvements and more complex residual structure. Notably, the theory's focus on local improvements offers a conservative perspective on capacity investments: even under random allocation ($R^2 = 0$), securing a modest screening capacity ($5-10\%$) is often the first priority, while at very high $R^2$, gains in policy value diminish so rapidly once $\alpha \geqslant \beta$ that the relative advantage of further prediction investments becomes negligible.

**When do small improvements in prediction error have the largest impact?** From theory (Theorem 3.2), we expect local policy value improvements from better predictions to diverge as $R^2 \to 0$ and $R^2 \to 1$ when $\alpha = \beta$. This aligns with our results in Figure 4: for small $\Delta_{R^2}$, the rate of local improvements in $V(R^2)$ with respect to $R^2$ diverges. The location of the maximum in $\alpha$ also follows from the theory: as $R^2 \to 1$, the rate only diverges for $\alpha = \beta$, while for small $R^2$ the maximum is at $\alpha \approx 0.5$.

**What are the relative benefits and tradeoffs of using a simpler vs more complex prediction model?** We compare a shallow 4-depth decision tree with the CatBoost model. As expected, the simpler tree shows a small drop in predictive power (5% decrease in $R^2$) which translates into a 1–8% reduction in policy value (see Figure 15). Compared to a uniform 5% increase in $R^2$ achieved by scaling the residuals (see Figure 14), the differences in policy value are only partially similar across $\alpha$. The CatBoost model does not provide a uniform improvement over the decision tree; for instance, it performs better at distinguishing longer unemployment spells.

Despite this performance gap, the simpler model offers potential advantages: it fits on a single sheet of paper, demands minimal computational infrastructure, can be easily explained to frontline case workers and resembles the categorical prioritization rules common in public institutions. (Johnson & Zhang, 2022). Because more complex models incur higher costs, a planner might instead increase screening capacity. Formally, we define

$$\Delta_\alpha^* = \inf_{\Delta_\alpha \in (0, 1-\beta)} \left\{ \Delta_\alpha : \frac{V_{\text{TREE}}(\alpha + \Delta_\alpha, \beta) - V_{\text{TREE}}(\alpha, \beta)}{V_{\text{CAT}}(\alpha, \beta) - V_{\text{TREE}}(\alpha, \beta)} \geqslant 1 \right\}$$

the smallest $\Delta_\alpha^*$ that matches the policy-value gains of the CatBoost model. Empirically, $\Delta_\alpha^*$ mostly rises with $\alpha$ (see Figure 15), consistent with our finding that the PAR decreases with $\alpha$. By framing the difference between models in terms of additional screenings, planners can directly compare the cost of increased capacity to that of deploying a more complex model.

## 6. Conclusion

This paper develops a framework for quantifying the relative value of prediction in identifying the worst-off. We formalize tradeoffs between expanding screening capacity and

---

[2] For the percentage of correctly identified LTU episodes, they report values of 0.29 at $\alpha \approx 0.1$ and 0.58 at $\alpha \approx 0.25$, compared to our observed values of 0.28 and 0.56, respectively.

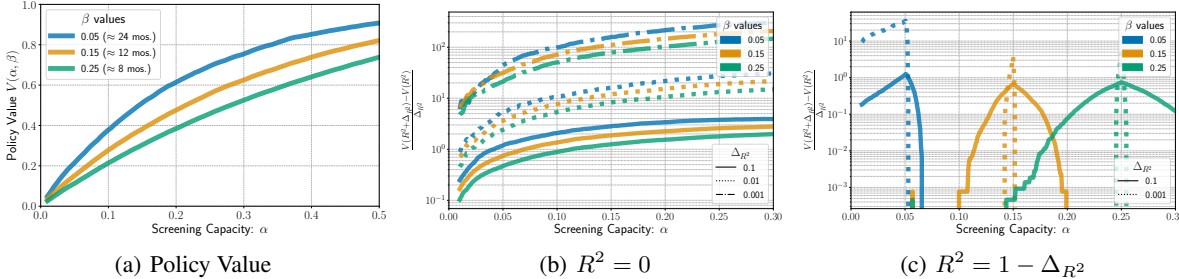

(a) Policy Value       (b) $R^2 = 0$       (c) $R^2 = 1 - \Delta_{R^2}$

*Figure 4.* (a) Policy value across different screening capacities ($\alpha$) and worst-off fractions $\beta$ evaluated on the test set using the `CatBoost` regression model. A $\beta$ value of 0.15 corresponds to the 12-month cutoff used to define long-term unemployment in Germany. (b, c) The rate of local improvements in $V(R^2)$ with respect to small changes in $R^2$. Panel (b) shows that the local improvements become increasingly large as $\Delta_{R^2}$ approaches zero. Panel (c) illustrates that improvements in prediction have the greatest impact when the capacity precisely matches the targeted fraction of the population ($\alpha = \beta$). Note that these are on a logarithmic scale.

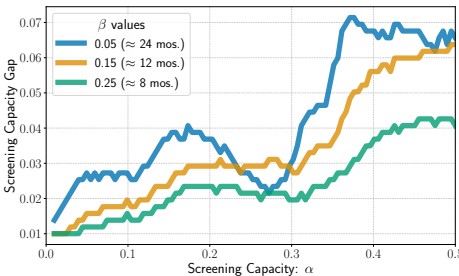

*Figure 5.* The minimum additional screening capacity that would need to be invested for the decision tree to achieve a policy value comparable to that of the `CatBoost` model.

improving predictive models, and show through both mathematical analysis and a real-world case study that prediction is not always the most important piece of the puzzle in social allocation systems. Future work could examine more specific application settings and cost structures, including distinctions between fixed and recurring costs, and explore policy levers that improve prediction unevenly, for example, by reducing errors in high-risk subgroups or by increasing robustness to distributional shifts. More broadly, we see a need for clearer theoretical foundations to understand the role of prediction in public-sector allocation, particularly in relation to the institutional and administrative systems in which it is embedded.

## Acknowledgements

This work is supported by the DAAD programme Konrad Zuse Schools of Excellence in Artificial Intelligence, sponsored by the Federal Ministry of Education and Research and by the Volkswagen Foundation, grant "Consequences of Artificial Intelligence for Urban Societies (CAIUS)". Juan Carlos Perdomo is supported by the Center for Research on Computation and Society (CRCS) at Harvard University and by the Alfred P. Sloan Foundation grant G-2020-13941. We would like to thank the anonymous reviewers for their insightful comments, as well as Frauke Kreuter, Patrick Schenk and Moritz Hardt for their valuable feedback.

## Impact Statement

Our work offers a principled framework for evaluating the relative benefits of using predictive models to target the most vulnerable populations, helping public agencies allocate limited resources more effectively. However, formalizing complex institutional processes inevitably omits important real-world details, risking biases or misalignments if assumptions are not carefully examined. We encourage policymakers and researchers to incorporate fairness, transparency, and accountability measures when implementing these methods, particularly in resource-constrained contexts where small design changes can disproportionately affect marginalized communities.

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

## A. Theoretical Investigation

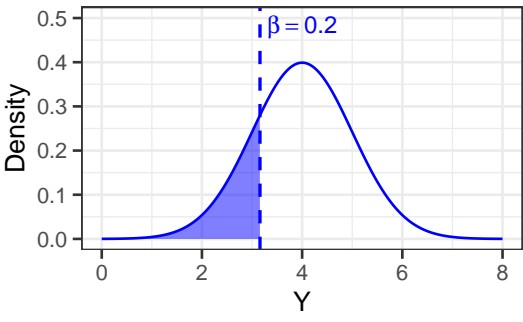

(a) Normal Welfare Distribution

*Figure 6.* Normal welfare distribution, with vertical lines marking the quantile cutoff ($\beta = 0.2$). The shaded region to the left of the vertical line represents the worst-off segment of the population.

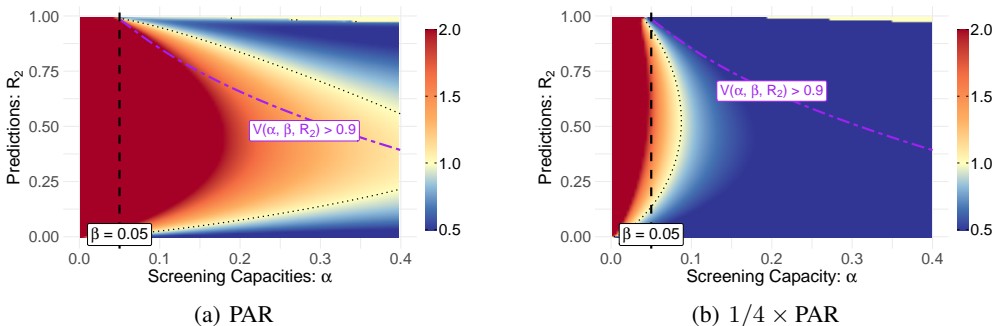

(a) PAR             (b) $1/4 \times$ PAR

*Figure 7.* **Numerical Simulation of the Prediction-Access Ratio (PAR)**, Equation 3, for $\Delta_{R^2} = \Delta_\alpha = 0.01$ and $\beta = 0.05$. Each point represents a screening capacity $\alpha$ (x-axis) and $R^2$ value (y-axis), with the color bar showing the PAR clipped to the range $[0.5, 2.0]$. The vertical black line marks $\beta$, indicating the threshold above which sufficient resources are available to screen everyone under perfect prediction. Dotted black lines represent PAR $= 1$, where improvements in $\alpha$ and $R^2$ are equally effective. The purple line marks the region in the $(\alpha, R^2)$ space where the policy value $V(\alpha, \beta, R^2)$ exceeds 0.9. Values above 0.9 are located in the upper-right region beyond the purple line.

## B. Experiments

### B.1. Experimental Setup and Labor Market Data

The dataset is provided via a Scientific Use File by the Research Data Centre (FDZ) of the German Federal Employment Agency (BA) at the Institute for Employment Research (IAB) (Schmucker & vom Berge, 2023a;b). It is a 2% weakly anonymized random sample of the complete German labor market records from 1975 to 2017 and contains information on 1,827,903 individuals across 62,340,521 observations (Schmucker & vom Berge, 2023b).

We follow the same set of covariates and aggregation procedure for individual unemployment spells as described in Bach et al. (2023), incorporating demographic characteristics, labor market histories, and information about the most recent job. This results in 56 numerical variables and 24 categorical variables, which are one-hot encoded for model training. Figure 8 shows a histogram of individual unemployment durations, which we use as the basis for constructing the outcome variables. The distribution is right-skewed, with a concentration on short durations near zero and a long tail. Such a pattern is commonly observed in other welfare-related outcomes, such as health or income metrics. We define as prediction target the duration of the unemployment period in days $Y$, capped at 24 months[3]. Differentiating tail values is less important for

---

[3]In practice, for a fixed $\beta$, the problem can also be framed as a classification task (see Appendix B.5).

decision-making, and capping also allows training across years with varying observation windows.

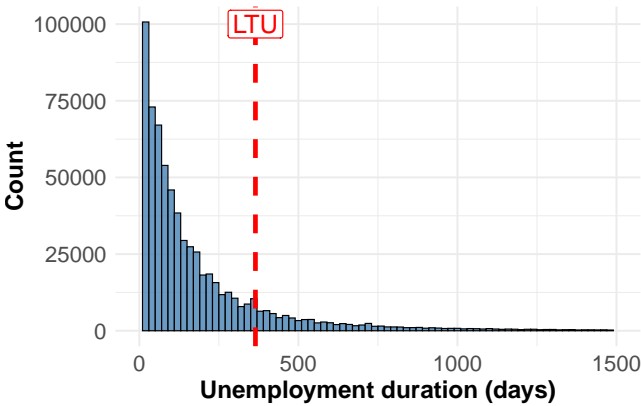

*Figure 8.* **Unemployment duration** The red line marks the 12 month threshold used to classify a jobseeking episode as long-term unemployment (LTU).

To avoid the impact of significant labor market reforms in Germany and to ensure full observation of unemployment durations up to 24 months, we restrict our analysis to unemployment episodes that began between 2010 and 2015. We use records from 2010 and 2011 to build the training dataset, records from 2012 for validation, and evaluate test performance on data from 2015 (see Figure 9). We left a gap between the training and test data periods to allow enough time for the outcomes in the training data to have been fully observed at test time, in order to mimic a realistic deployment scenario starting at the beginning of 2015.

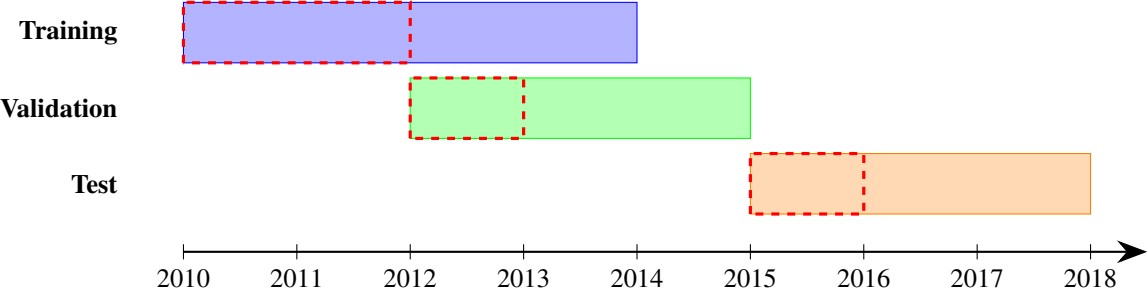

*Figure 9.* Stacked timeline diagram illustrating training (2010–2013), validation (2012–2014), and test (2015–2017) data periods. Red dashed boundaries within each colored box indicate the possible start dates of unemployment episodes, while the full colored boxes represent the entire observation phases for each dataset.

### B.2. Training Details

We use CatBoost (https://catboost.ai) for model training. The model was trained for a maximum 5,000 iterations with an early stopping criterion (early_stopping_rounds = 20) based on validation performance. Additionally, we train a shallow Decision Tree (max_depth = 4) using the scikit-learn package. All hyperparameters are kept at their default settings unless otherwise specified.

### B.3. Prediction Improvements

To simulate an increase in predictive power by a specified amount $\Delta_{R^2}$, we adjust the model's predictions $\hat{Y}$ using the residuals $Y - \hat{Y}$. Starting with the original predictions $\hat{Y}$ and true outcomes $Y$, we define the adjusted predictions as

$$\hat{Y}_+ = \hat{Y} + \delta(Y - \hat{Y})$$

We can then determine the $\delta$ corresponding to an increase of $\Delta_{R^2}$ in the model's $R^2$:

$$\delta = 1 - \sqrt{1 - \Delta_{R^2} \frac{\sum_{i=1}^{n}(Y_i - \bar{Y})^2}{\sum_{i=1}^{n}(Y_i - \hat{Y}_i)^2}}$$

For a specified $\delta$, the new residuals are

$$Y - \hat{Y}_+ = (1 - \delta)(Y - \hat{Y})$$

Consequently, the variance decreases by a multiplicative factor: $\text{Var}(Y - \hat{Y}_+) = (1 - \delta)^2 \text{Var}(Y - \hat{Y})$.

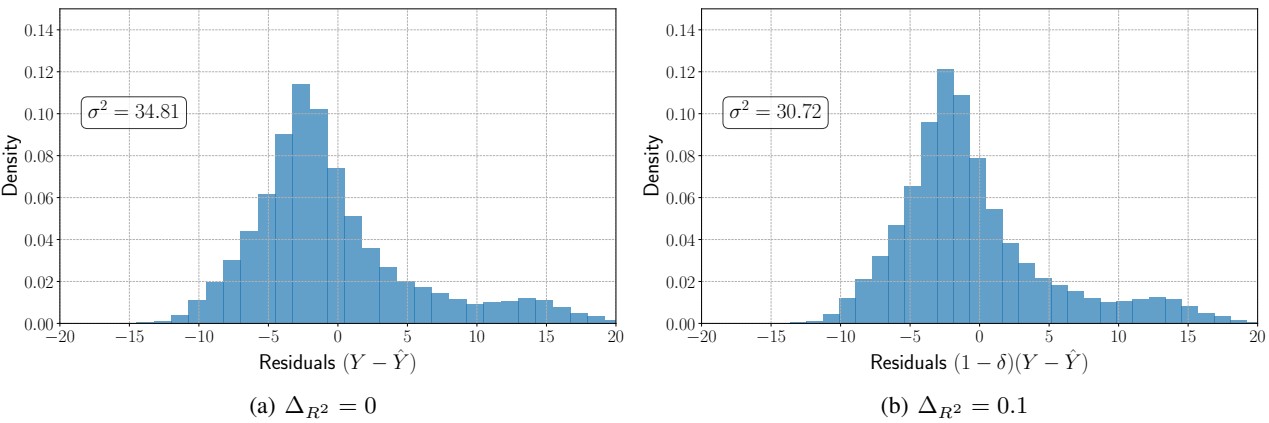

(a) $\Delta_{R^2} = 0$             (b) $\Delta_{R^2} = 0.1$

*Figure 10.* **Residual Distribution Before and After Adjustment** Figure (a) shows the residual distribution for the original predictions ($\Delta_{R^2} = 0$), while Figure (b) shows the residual distribution after increasing the $R^2$-value ($\Delta_{R^2} = 0.1$) for the CatBoost model. The adjustment preserves the overall structure of the residuals.

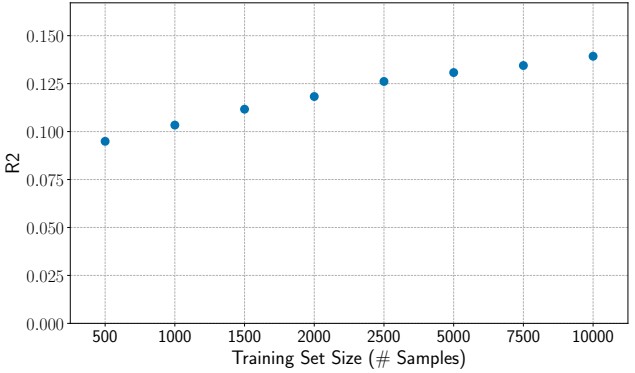

*Figure 11.* The $R^2$ value on the test set for varying training set size (CatBoost Regression).

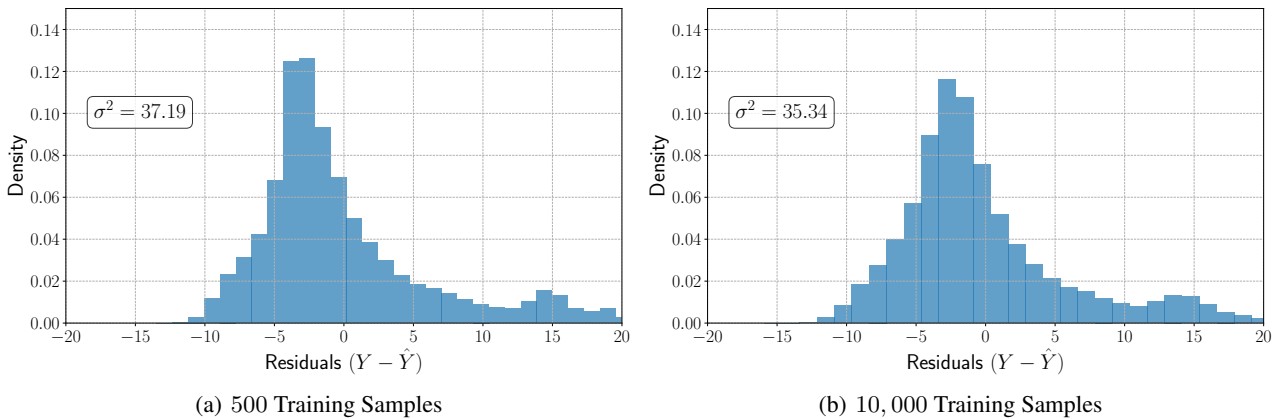

(a) 500 Training Samples          (b) 10,000 Training Samples

*Figure 12.* Residual distributions on the test set for models trained with varying training set sizes.

## B.4. Additional Figures

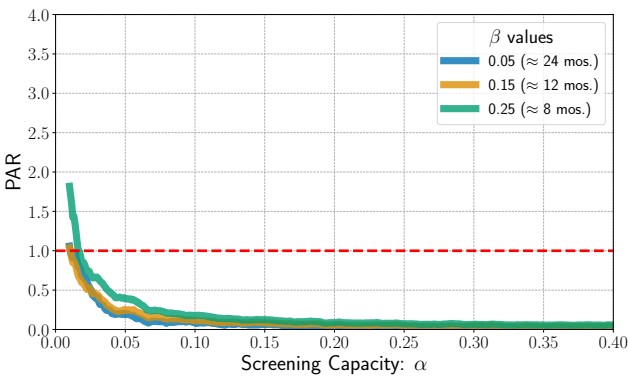

*Figure 13.* Prediction-Access Ratio for $R^2 = 0$ and $\Delta_{R^2} = \Delta_\alpha = 0.01$.

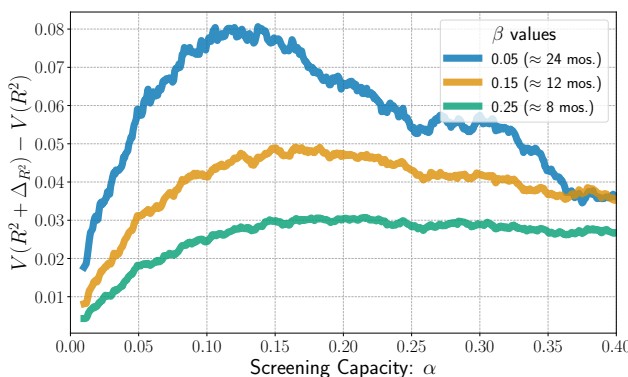

*Figure 14.* $V(R^2 + \Delta_{R^2}) - V(R^2)$ for `CatBoost` model and $\Delta_{R^2} = 0.05$.

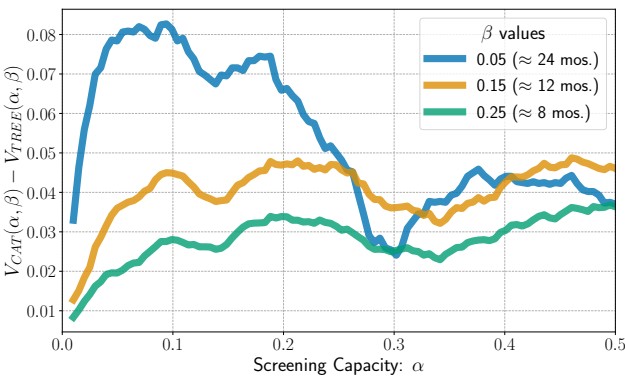

*Figure 15.* The difference in policy value between a 4-depth decision tree and `CatBoost` model.

## B.5. Binary Classification

Instead of predicting the exact duration of unemployment, the problem can be reframed as a binary classification task. For a fixed $\beta$, we can define a binary outcome: $Y = 1\{Y \geqslant F_{Y,n}^{-1}(1 - \beta)\}$. This approach more directly encodes the target of interest: identifying individuals who may require further screening or assistance. If the chosen classifier provides estimates of class probabilities $\hat{p}(x)$, it can be used to formulate a decision policy $1\{\hat{p}(x) \geqslant F_{n,\hat{p}}^{-1}(1 - \alpha)\}$. However, this forces us to specify $\beta$ and the resulting decision threshold prior to model training. This requirement reduces flexibility compared to a continuous prediction setup, making classification more appropriate when the model is not intended for use in other tasks and when $\beta$ remains constant across the deployment context. Additionally, directly converting durations to labels discards information on the precise unemployment durations that could be valuable for the modeling process.

As can be seen in Figure 16, the resulting policy values and true positive counts remain very similar compared to the regression case.

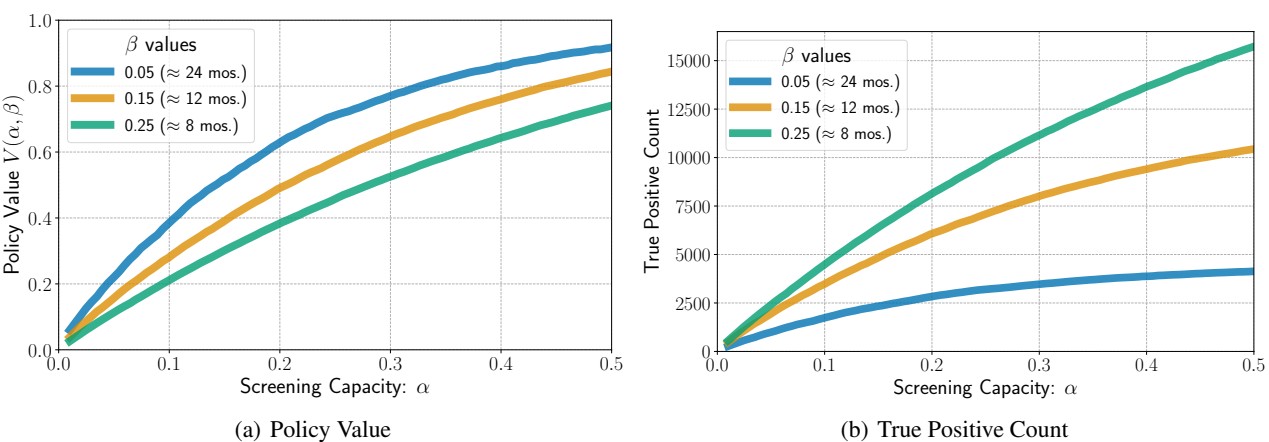

(a) Policy Value

(b) True Positive Count

*Figure 16.* Policy Value and True Positive Count on Test Set (Classification).

## C. Additional Propositions

**Proposition C.1.** *(Optimal Policy with Gaussian Error) If $\varepsilon = Y - \hat{Y} \sim \mathcal{N}(0, \gamma^2)$, then the optimal policy $\pi^* : \mathbb{R} \to \{0, 1\}$ to solve the screening problem (Definition 2.1) is equal to:*

$$\pi^*(\hat{Y}_i) = 1\{\hat{Y}_i \leqslant F_{\hat{Y}}^{-1}(\alpha)\}$$

*where $F_{\hat{Y}}^{-1}(\alpha)$ is the $\alpha$-quantile of $\hat{Y}$. The value of the policy is $V(\pi^*) = \Pr[\hat{Y} \leqslant F_{\hat{Y}}^{-1}(\alpha) \mid Y \leqslant F_Y^{-1}(\beta)]$.*

*Proof.* Since $Y = \hat{Y} + \varepsilon$ where $\varepsilon \sim \mathcal{N}(0, \gamma^2)$, it follows for the conditional distribution $Y \mid \hat{Y} \sim \mathcal{N}(\hat{Y}, \gamma^2)$. Since $Y \mid \hat{Y}$ is Gaussian, we can express the conditional probability from Proposition 2.2 in terms of the CDF of the standard normal distribution,

$$\Pr[Y \leqslant F_Y^{-1}(\beta) \mid \hat{Y}] = \Phi\left(\frac{F_Y^{-1}(\beta) - \hat{Y}}{\gamma}\right)$$

To reproduce the ranking induced by $\Pr[Y \leqslant F_Y^{-1}(\beta) \mid \hat{Y}]$, individuals can be ranked in ascending order of $\hat{Y}$. Thus, we can express the optimal policy (Proposition 2.2) in terms of a ranking of $\hat{Y}$,

$$\pi^*(\hat{Y}_i) = 1\{\hat{Y}_i \leqslant F_{\hat{Y}}^{-1}(\alpha)\}$$

where $F_{\hat{Y}}^{-1}(\alpha)$ is the $\alpha$-quantile of $\hat{Y}$. The value $V(\pi^*)$ that can by achieved by the optimal screening policy $\pi^*$ can then be expressed as:

$$V(\pi^*) = \mathbb{E}\left[\pi^*(\hat{Y}) = 1 \mid Y \leqslant F_Y^{-1}(\beta)\right] = \mathbb{E}\left[1\{\hat{Y} \leqslant F_{\hat{Y}}^{-1}(\alpha)\} \mid Y \leqslant F_Y^{-1}(\beta)\right]$$
$$= \Pr[\hat{Y} \leqslant F_{\hat{Y}}^{-1}(\alpha) \mid Y \leqslant F_Y^{-1}(\beta)]$$

$\blacksquare$

## D. Proofs

### D.1. Optimal Policy for Screening Problem: Proof of Proposition 2.2

*Proof.* We rewrite the policy value,

$$\mathbb{E}\left[\pi(\hat{Y}_i) = 1 \mid Y \leqslant F_Y^{-1}(\beta)\right] = \frac{\mathbb{E}\left[\pi(\hat{Y}_i)1\{Y \leqslant F_Y^{-1}(\beta)\}\right]}{\Pr[Y \leqslant F_Y^{-1}(\beta)]}$$
$$= \frac{1}{\beta}\mathbb{E}\left[\pi(\hat{Y}_i)\mathbb{E}\left[1\{Y \leqslant F_Y^{-1}(\beta)\} \mid \hat{Y}_i\right]\right]$$
$$= \frac{1}{\beta}\mathbb{E}\left[\pi(\hat{Y}_i)\Pr[Y \leqslant F_Y^{-1}(\beta) \mid \hat{Y}_i]\right]$$

To maximize the objective, individuals $\hat{Y}_i$ with the largest scores $s(\hat{Y}_i) = \Pr[Y \leqslant F_Y^{-1}(\beta) \mid \hat{Y}_i]$ should be prioritized. Thus, the optimal policy is to intervene ($\pi(\hat{Y}_i) = 1$) for the top $\alpha$-fraction of the population ranked by $\Pr[Y \leqslant F_Y^{-1}(\beta) \mid \hat{Y}]$. $\blacksquare$

### D.2. Optimal Policy Value in Gaussian Setting: Proof of Proposition 2.3

Following Proposition C.1, the value of the optimal screening policy $\pi^*$ can then be expressed as:

$$V(\pi^*) = \Pr[\hat{Y} \leqslant F_{\hat{Y}}^{-1}(\alpha) \mid Y \leqslant F_Y^{-1}(\beta)]$$

We can rewrite the conditional probability in terms of the joint distribution of $Y$ and $\hat{Y}$, and note that $\Pr\{Y \leqslant F_Y^{-1}(\beta)\} = F_Y(F_Y^{-1}(\beta)) = \beta$,

$$\Pr[\hat{Y} \leqslant F_{\hat{Y}}^{-1}(\alpha) \mid Y \leqslant F_Y^{-1}(\beta)] = \frac{1}{\beta}\Pr[\hat{Y} \leqslant F_{\hat{Y}}^{-1}(\alpha), Y \leqslant F_Y^{-1}(\beta)]$$

We then standardize $Y \sim \mathcal{N}(\mu, \eta^2)$ and $\hat{Y} \sim \mathcal{N}(\mu, \eta^2 - \gamma^2)$ and make use that for a normal random variable with mean $\mu$ and variance $\sigma^2$ the quantile function is $F^{-1}(p) = \mu + \sigma \Phi^{-1}(p)$.

$$\frac{1}{\beta} \Pr[\hat{Y} \leqslant F_{\hat{Y}}^{-1}(\alpha), Y \leqslant F_Y^{-1}(\beta)] = \frac{\Pr\left\{ Z_1 \leqslant \frac{F_{\hat{Y}}^{-1}(\alpha) - \mu}{\sqrt{\eta^2 - \gamma^2}}, Z_2 \leqslant \frac{F_Y^{-1}(\beta) - \mu}{\eta} \right\}}{\beta}$$

$$= \frac{\Pr\left\{ Z_1 \leqslant \Phi^{-1}(\alpha), Z_2 \leqslant \Phi^{-1}(\beta) \right\}}{\beta}$$

$Z_1$ and $Z_2$ are standard Gaussian with $\mathrm{Cov}(Z_1, Z_2) = \mathbb{E}[Z_1 Z_2] = \frac{1}{\eta \sqrt{\eta^2 - \gamma^2}} \mathrm{Cov}(\hat{Y}, \hat{Y} + \varepsilon) = \frac{\mathrm{Cov}(\hat{Y}, \hat{Y})}{\eta \sqrt{\eta^2 - \gamma^2}} = \frac{\sqrt{\eta^2 - \gamma^2}}{\eta}$. Thus,

they are distributed according to a standard bivariate normal distribution with correlation $\rho = \mathrm{Cov}(Z_1, Z_2) = \frac{\sqrt{\eta^2 - \gamma^2}}{\eta}$.
Thus,

$$V(\pi^*) = \mathbb{E}\left[ \pi^*(\hat{Y}) = 1 \mid Y \leqslant F_Y^{-1}(\beta) \right] = \frac{1}{\beta} \Phi_2\left( \Phi^{-1}(\alpha), \Phi^{-1}(\beta); \rho \right)$$

where

$$\Phi_2\left( \Phi^{-1}(\alpha), \Phi^{-1}(\beta) \right) = \int_{-\infty}^{\Phi^{-1}(\alpha)} \int_{-\infty}^{\Phi^{-1}(\beta)} \phi_2(z_1, z_2; \rho) \, \mathrm{d}z_1 \, \mathrm{d}z_2$$

and

$$\phi_2(z_1, z_2) = \frac{1}{2\pi \sqrt{1 - \rho^2}} e^{-1/2(z_1^2 - 2\rho z_1 z_2 + z_2^2)/(1 - \rho^2)}$$

### D.3. Prediction-Access Ratio for Small Screening Capacities: Proof of Theorem 3.1

Using Taylor's theorem,

$$V(\alpha, \beta, R^2 + \Delta_{R^2}) - V(\alpha, \beta, R^2) = \Delta_{R^2} \frac{\partial}{\partial R^2} V(\alpha, \beta, R^2 + p_{R^2} \Delta_{R^2})$$

where $p_{R^2} \in (0, 1)$. We know from Lemma D.3,

$$\frac{\partial}{\partial R^2} V(\alpha, \beta, R_*^2) \leqslant \frac{1}{\beta \sqrt{8\pi R_*^2 (1 - R_*^2)}} \phi\left( \frac{\Phi^{-1}(\alpha) - \sqrt{R_*^2} \Phi^{-1}(\beta)}{\sqrt{1 - R_*^2}} \right)$$

where $R_*^2 := R^2 + p_{R^2} \Delta_{R^2}$. For $\alpha < 0.5$ and $\beta \leqslant 0.5$, we know $\Phi^{-1}(\alpha) < 0$ and $\Phi^{-1}(\beta) \leqslant 0$. It follows, that for any $\varepsilon_1 > 0$, $0 < R_*^2$ and $0 < \beta$, there exists a threshold value $t_1 > 0$, such that for all $\alpha \leqslant t_1$, we have

$$(1 + \varepsilon_1) \frac{\Phi^{-1}(\alpha)}{\sqrt{1 - R_*^2}} \leqslant \frac{\Phi^{-1}(\alpha) - \sqrt{R_*^2} \Phi^{-1}(\beta)}{\sqrt{1 - R_*^2}} \leqslant (1 - \varepsilon_1) \frac{\Phi^{-1}(\alpha)}{\sqrt{1 - R_*^2}}$$

If $\alpha < \beta$ we find $\Phi^{-1}(\alpha) - \sqrt{R_*^2} \Phi^{-1}(\beta) < 0$. Since $\phi(x) \leqslant \phi(x')$ for $x \leqslant x' < 0$,

$$\frac{1}{\beta \sqrt{8\pi R_*^2 (1 - R_*^2)}} \phi\left( \frac{\Phi^{-1}(\alpha) - \sqrt{R_*^2} \Phi^{-1}(\beta)}{\sqrt{1 - R_*^2}} \right) \leqslant \frac{1}{\beta \sqrt{8\pi R_*^2 (1 - R_*^2)}} \phi\left( (1 - \varepsilon_1) \frac{\Phi^{-1}(\alpha)}{\sqrt{1 - R_*^2}} \right)$$

$$= \frac{1}{\beta \sqrt{8\pi R_*^2 (1 - R_*^2)}} \phi\left( \kappa \Phi^{-1}(\alpha) \right)$$

$$= \frac{1}{\beta \sqrt{8\pi R_*^2 (1 - R_*^2)}} \phi\left( \kappa \Phi^{-1}(1 - \alpha) \right)$$

where $\kappa := \frac{(1-\varepsilon_1)}{\sqrt{1-R_*^2}}$. For any $\varepsilon_2 > 0$, there exists a threshold $t_2 > 0$, such that for all $\alpha \leqslant t_2$, we can apply *Lemma B.5.* from Perdomo (2024) to arrive at the following inequality:

$$\phi\left(\kappa\Phi^{-1}(1-\alpha)\right) \leqslant \frac{1}{\sqrt{2\pi}}\left((1+\varepsilon_2)\sqrt{2\pi}\alpha\Phi^{-1}(1-\alpha)\right)^{\kappa^2}$$

Thus,

$$V(\alpha,\beta,R^2+\Delta_{R^2}) - V(\alpha,\beta,R^2) \leqslant \Delta_{R^2}\frac{1}{\beta 4\pi\sqrt{R_*^2(1-R_*^2)}}\left((1+\varepsilon_2)\sqrt{2\pi}\alpha\Phi^{-1}(1-\alpha)\right)^{\kappa^2}$$

We can use Taylor's theorem again and from Lemma D.1 we know that

$$V(\alpha+\Delta_\alpha,\beta,R^2) - V(\alpha,\beta,R^2) = \Delta_\alpha\frac{\partial}{\partial\alpha}V(\alpha+p_\alpha\Delta_\alpha,\beta,R^2)$$
$$= \Delta_\alpha\frac{1}{\beta}\Phi\left(\frac{\Phi^{-1}(\beta)-\sqrt{R^2}\Phi^{-1}(\alpha+p_\alpha\Delta_\alpha)}{\sqrt{1-R^2}}\right)$$

where $p_\alpha \in (0,1)$. Since $0 < \beta$ and $0 < R^2$ there will always be a small enough $\alpha + \Delta_\alpha$ such that

$$\Phi^{-1}(\beta) - \sqrt{R^2}\Phi^{-1}(\alpha+p_\alpha\Delta_\alpha) \geqslant 0$$

Since $\Phi(x) \geqslant 1/2$ for $x \geqslant 0$, it follows

$$\frac{\Delta_\alpha}{2\beta} \leqslant V(\alpha+\Delta_\alpha,\beta,R^2) - V(\alpha,\beta,R^2)$$

It follows for the prediction-access ratio,

$$\frac{\Delta_\alpha}{\Delta_{R^2}}2\pi\sqrt{R_*^2(1-R_*^2)}\left((1+\varepsilon_2)\sqrt{2\pi}\alpha\Phi^{-1}(1-\alpha)\right)^{-(1-\varepsilon_1)^2\frac{1}{1-R_*^2}} \leqslant \frac{V(\alpha+\Delta_\alpha,\beta,R^2) - V(\alpha,\beta,R^2)}{V(\alpha,\beta,R^2+\Delta_{R^2}) - V(\alpha,\beta,R^2)}$$

For small $\alpha$, $\Phi^{-1}(1-\alpha)$ grows asymptotically like $\sqrt{\log(1/\alpha)}$. Consequently, the polynomial growth of $\alpha^{-1/(1-R^2)}$ drives the PAR to increase rapidly as $\alpha$ decreases. Since $\frac{1}{1-R_*^2}$ increases with $R_*^2$ and $R^2 \leqslant R_*^2$, we can lower bound the PAR by inserting $R^2$ instead of $R_*^2$:

$$\frac{\Delta_\alpha}{\Delta_{R^2}}2\pi\sqrt{R^2(1-R^2)}\left((1+\varepsilon_2)\sqrt{2\pi}\alpha\Phi^{-1}(1-\alpha)\right)^{-(1-\varepsilon_1)^2\frac{1}{1-R^2}} \leqslant \frac{V(\alpha+\Delta_\alpha,\beta,R^2) - V(\alpha,\beta,R^2)}{V(\alpha,\beta,R^2+\Delta_{R^2}) - V(\alpha,\beta,R^2)}$$

We can simplify the lower-bound by noting that $0 < \varepsilon_1$ and $0 < \varepsilon_2$ can be made arbitrarily small by selecting a sufficiently small threshold $t$ for $\alpha + \Delta_\alpha$. Specifically, $\varepsilon_2 < 1$ holds for $\alpha \leqslant 0.05$ (see *Lemma A.6* in Perdomo (2024)).

$$\frac{\Delta_\alpha}{\Delta_{R^2}}\sqrt{R^2(1-R^2)}\left(5.1\cdot\alpha\Phi^{-1}(1-\alpha)\right)^{-\frac{1}{1-R^2}+o(1)} \leqslant \frac{V(\alpha+\Delta_\alpha,\beta,R^2) - V(\alpha,\beta,R^2)}{V(\alpha,\beta,R^2+\Delta_{R^2}) - V(\alpha,\beta,R^2)}$$

### D.4. Maximally Effective (Local) Prediction Improvements: Proof of Theorem 3.2

We know from Lemma D.2,

$$\lim_{\Delta\to 0}\frac{V(\alpha,\beta,R^2+\Delta) - V(\alpha,\beta,R^2)}{\Delta} = \frac{\partial}{\partial R^2}V(\alpha,\beta,R^2)$$
$$= \frac{1}{2\beta\sqrt{R^2}}\phi_2\left(\Phi^{-1}(\alpha),\Phi^{-1}(\beta);\rho\right)$$

We insert $\phi_2\left(\cdot\right)$ and arrive at

$$\frac{\partial}{\partial R^2}V(\alpha,\beta,R^2) = \underbrace{\frac{1}{4\pi\beta\sqrt{R^2(1-R^2)}}}_{T_1}$$

$$\times \underbrace{\exp\left(-\frac{1}{2(1-R^2)}(\Phi^{-1}\left(\alpha\right)^2 + \Phi^{-1}\left(\beta\right)^2 - 2\sqrt{R^2}\Phi^{-1}\left(\alpha\right)\Phi^{-1}\left(\beta\right))\right)}_{T_2}$$

The prefactor $T_1$ diverges as $R^2 \to 1$ or $R^2 \to 0$.

If $R^2 \to 1$, the exponential term will generally suppress the polynomial growth of the prefactor. However for $\alpha = \beta$, we find for the exponent

$$-\frac{1}{2(1-R^2)}(\Phi^{-1}\left(\alpha\right)^2 + \Phi^{-1}\left(\beta\right)^2 - 2\sqrt{R^2}\Phi^{-1}\left(\alpha\right)\Phi^{-1}\left(\beta\right)) = -\frac{1-\sqrt{R^2}}{1-R^2}\Phi^{-1}(\beta)^2$$

$$= -\frac{1}{(1+\sqrt{R^2})}\Phi^{-1}\left(\alpha\right)^2$$

$$\stackrel{R^2\to 1}{=} -\frac{1}{2}\Phi^{-1}\left(\beta\right)^2$$

which is finite. Therefore, $\frac{\partial}{\partial R^2}V(\alpha,\beta,R^2)$ becomes unboundedly large if $\alpha = \beta$ and $R^2 \to 1$.

If $R^2 \to 0$, the prefector $T_1$ diverges again to $+\infty$. The exponent then simplifies to

$$-\frac{1}{2(1-R^2)}(\Phi^{-1}\left(\alpha\right)^2 + \Phi^{-1}\left(\beta\right)^2 - 2\sqrt{R^2}\Phi^{-1}\left(\alpha\right)\Phi^{-1}\left(\beta\right)) = -\frac{1}{2}(\Phi^{-1}\left(\alpha\right)^2 + \Phi^{-1}\left(\beta\right)^2)$$

If $\alpha$ and $\beta$ are not set arbitrarily small or large $\frac{\partial}{\partial R^2}V(\alpha,\beta,R^2)$ will diverge. The local PAR (Lemma D.1)

$$\lim_{\Delta\to 0}\frac{V(\alpha+\Delta,\beta,R^2) - V(\alpha,\beta,R^2)}{V(\alpha,\beta,R^2+\Delta) - V(\alpha,\beta,R^2)} = \frac{\frac{\partial}{\partial\alpha}V(\alpha,\beta,R^2)}{\frac{\partial}{\partial R^2}V(\alpha,\beta,R^2)}$$

$$= \frac{\Phi\left(\frac{\Phi^{-1}(\beta)-\sqrt{R^2}\Phi^{-1}(\alpha)}{\sqrt{1-R^2}}\right)}{\frac{1}{2\sqrt{R^2}}\phi_2\left(\Phi^{-1}\left(\alpha\right),\Phi^{-1}\left(\beta\right);\rho\right)}$$

approaches zero in both regimes.

### D.5. Prediction-Access Ratio for Local Improvements: Proof of Proposition 3.3

We know

$$\lim_{\Delta\to 0}\frac{V(\alpha+\Delta,\beta,R^2) - V(\alpha,\beta,R^2)}{V(\alpha,\beta,R^2+\Delta) - V(\alpha,\beta,R^2)} = \frac{\frac{\partial}{\partial\alpha}V(\alpha,\beta,R^2)}{\frac{\partial}{\partial R^2}V(\alpha,\beta,R^2)}$$

Using Lemma D.1 and Lemma D.3 we find a lower bound for the PAR:

$$\underbrace{\sqrt{8\pi R^2(1-R^2)}}_{T_1}\underbrace{\frac{\Phi\left(\frac{\Phi^{-1}(\beta)-\sqrt{R^2}\Phi^{-1}(\alpha)}{\sqrt{1-R^2}}\right)}{\phi\left(\frac{\Phi^{-1}(\beta)-\sqrt{R^2}\Phi^{-1}(\alpha)}{\sqrt{1-R^2}}\right)}}_{T_2} \leqslant \frac{V(\alpha+\Delta_\alpha,\beta,R^2) - V(\alpha,\beta,R^2)}{V(\alpha,\beta,R^2+\Delta_{R^2}) - V(\alpha,\beta,R^2)}$$

We then denote $z := \frac{\Phi^{-1}(\beta)-\sqrt{R^2}\Phi^{-1}(\alpha)}{\sqrt{1-R^2}}$ and $T_2 = \frac{\Phi(z)}{\phi(z)}$. We know from Lemma D.4 that $\frac{\Phi(z)}{\phi(z)}$ increases with $z$. It follows that we need to find the smallest possible $z$ to find a lower bound for $T_2$. Generally, $z$ decreases with $\alpha$ and increases with $\beta$. We treat both cases separately:

1. For $\alpha \leqslant \beta$ we find $\frac{\Phi^{-1}(\beta)(1-\sqrt{R^2})}{\sqrt{1-R^2}} \leqslant z$. Since $\frac{1-\sqrt{R^2}}{\sqrt{1-R^2}}$ decreases with $R^2$ and $\beta \leqslant 0.5$ we can lower bound the expression by setting $R^2 = 0.15$ and $\beta = 0.03$. Thus, $-1.25 \leqslant z$ and $0.59 \leqslant T_2$. Since $0.15 \leqslant R^2 \leqslant 0.85$ we can lower bound the prefactor $1.79 \leqslant T_1$.

2. For $\alpha \leqslant 0.5$, it follows $\frac{\Phi^{-1}(\beta)}{\sqrt{1-R^2}} \leqslant z$ by setting $\Phi^{-1}(\alpha = 0.5) = 0$. Since $0.15 \leqslant \beta$ and $0.2 \leqslant R^2 \leqslant 0.5$, it follows $0.52 \leqslant T_2$ and $2 \leqslant T_1$

In both cases, we can combine the lower bounds of $T_1$ and $T_2$ to find

$$1 \leqslant \frac{V(\alpha + \Delta_\alpha, \beta, R^2) - V(\alpha, \beta, R^2)}{V(\alpha, \beta, R^2 + \Delta_{R^2}) - V(\alpha, \beta, R^2)}$$

### D.6. Technical Lemmas

**Lemma D.1** (Derivative w.r.t. $\alpha$).

$$\frac{\partial}{\partial \alpha} V(\alpha, \beta, R^2) = \frac{1}{\beta} \Phi \left( \frac{\Phi^{-1}(\beta) - \sqrt{R^2}\Phi^{-1}(\alpha)}{\sqrt{1-R^2}} \right) \tag{4}$$

*Proof.* In the Gaussian setting we find for the policy value (Proposition 2.3),

$$V(\alpha, \beta, R^2) = \frac{\Phi_2\left(\Phi^{-1}(\alpha), \Phi^{-1}(\beta); \rho\right)}{\beta}$$

We first apply Leibniz integral rule,

$$\frac{\partial}{\partial \alpha} V(\alpha, \beta, R^2) = \frac{\partial}{\partial \alpha} \frac{\Phi_2\left(\Phi^{-1}(\alpha), \Phi^{-1}(\beta); \rho\right)}{\beta}$$

$$= \frac{1}{\beta} \int_{-\infty}^{\Phi^{-1}(\beta)} \phi_2\left(z_1, \Phi^{-1}(\alpha); \rho\right) \, \mathrm{d}z_1 \frac{\partial}{\partial \alpha} \Phi^{-1}(\alpha)$$

$$= \frac{1}{\beta \phi\left(\Phi^{-1}(\alpha)\right)} \int_{-\infty}^{\Phi^{-1}(\beta)} \phi_2\left(\Phi^{-1}(\alpha), z_2; \rho\right) \, \mathrm{d}z_2$$

We insert the bivariate density $\phi_2(\cdot)$ and substitute $z_2 - \rho\Phi^{-1}(\alpha) = u\sqrt{1-\rho^2}$

$$\frac{1}{\beta \phi\left(\Phi^{-1}(\alpha)\right)} \int_{-\infty}^{\Phi^{-1}(\beta)} \phi_2\left(\Phi^{-1}(\alpha), z_2; \rho\right) \, \mathrm{d}z_2$$

$$= \frac{1}{\beta \phi\left(\Phi^{-1}(\alpha)\right)} \frac{1}{2\pi\sqrt{1-\rho^2}} \int_{-\infty}^{\Phi^{-1}(\beta)} e^{-1/2(z_2^2 - 2\rho z_2 \Phi^{-1}(\alpha) + \Phi^{-1}(\alpha)^2)/(1-\rho^2)} \, \mathrm{d}z_2$$

$$= \frac{1}{2\pi\beta \phi\left(\Phi^{-1}(\alpha)\right)} \int_{-\infty}^{(\Phi^{-1}(\beta) - \rho\Phi^{-1}(\alpha))/\sqrt{1-\rho^2}} e^{-1/2(u^2(1-\rho^2) + \rho^2\Phi^{-1}(\alpha)^2 + (1-\rho^2)\Phi^{-1}(\alpha)^2)/(1-\rho^2)} \, \mathrm{d}u$$

$$= \frac{1}{2\pi\beta \phi\left(\Phi^{-1}(\alpha)\right)} e^{-1/2\Phi^{-1}(\alpha)^2} \int_{-\infty}^{(\Phi^{-1}(\beta) - \rho\Phi^{-1}(\alpha))/\sqrt{1-\rho^2}} e^{-1/2u^2} \, \mathrm{d}u$$

$$= \frac{1}{\beta} \Phi \left( \frac{\Phi^{-1}(\beta) - \rho\Phi^{-1}(\alpha)}{\sqrt{1-\rho^2}} \right) = \frac{1}{\beta} \Phi \left( \frac{\Phi^{-1}(\beta) - \sqrt{R^2}\Phi^{-1}(\alpha)}{\sqrt{1-R^2}} \right)$$

$\blacksquare$

**Lemma D.2** (Derivative w.r.t. $R^2$).

$$\frac{\partial}{\partial R^2} V(\alpha, \beta, R^2) = \frac{1}{2\beta\sqrt{R^2}} \phi_2\left(\Phi^{-1}(\alpha), \Phi^{-1}(\beta)\right) \tag{5}$$

*where $\phi_2(\cdot)$ is the standard bivariate density.*

*Proof.*

$$
\begin{aligned}
\frac{\partial}{\partial R^2} V(\alpha, \beta, R^2) &= \frac{\partial}{\partial R^2} \frac{\Phi_2\left(\Phi^{-1}(\alpha), \Phi^{-1}(\beta); \rho\right)}{\beta} \\
&= \frac{1}{\beta} \frac{\partial \rho}{\partial R^2} \frac{\partial}{\partial \rho} \Phi_2\left(\Phi^{-1}(\alpha), \Phi^{-1}(\beta); \rho\right) \\
&= \frac{1}{2\beta\sqrt{R^2}} \frac{\partial}{\partial \rho} \Phi_2\left(\Phi^{-1}(\alpha), \Phi^{-1}(\beta); \rho\right) \\
&= \frac{1}{2\beta\sqrt{R^2}} \phi_2\left(\Phi^{-1}(\alpha), \Phi^{-1}(\beta); \rho\right)
\end{aligned}
$$

where $\phi_2(\cdot)$ is the standard bivariate density. We utilized $R^2 = \rho^2$, and in the final step applied the partial derivative of the standard bivariate cumulative distribution with respect to its correlation $\rho$ (Drezner & Wesolowsky, 1990). ∎

**Lemma D.3** (Upper bound of $R^2$ derivative). *Let $0 \leqslant \sqrt{R^2} \leqslant 1$. Then,*

$$
\frac{\partial}{\partial R^2} V(\alpha, \beta, R^2) \leqslant \frac{1}{\beta\sqrt{8\pi R^2(1-R^2)}} \phi\left(\frac{\Phi^{-1}(\beta) - \sqrt{R^2}\Phi^{-1}(\alpha)}{\sqrt{1-R^2}}\right) \tag{6}
$$

$$
\frac{\partial}{\partial R^2} V(\alpha, \beta, R^2) \leqslant \frac{1}{\beta\sqrt{8\pi R^2(1-R^2)}} \phi\left(\frac{\sqrt{R^2}\Phi^{-1}(\beta) - \Phi^{-1}(\alpha)}{\sqrt{1-R^2}}\right) \tag{7}
$$

*Proof.* We know from Lemma D.2,

$$
\begin{aligned}
\frac{\partial}{\partial R^2} V(\alpha, \beta, R^2) &= \frac{1}{2\beta\sqrt{R^2}} \phi_2\left(\Phi^{-1}(\alpha), \Phi^{-1}(\beta); \rho\right) \\
&= \frac{1}{4\pi\beta\sqrt{R^2(1-R^2)}} \\
&\quad \times \exp\left(-\frac{1}{2(1-R^2)}(\Phi^{-1}(\alpha)^2 + \Phi^{-1}(\beta)^2 - 2\sqrt{R^2}\Phi^{-1}(\alpha)\Phi^{-1}(\beta))\right)
\end{aligned}
$$

Since $0 \leqslant \sqrt{R^2} \leqslant 1$,

$$
\begin{aligned}
\Phi^{-1}(\alpha)^2 + \Phi^{-1}(\beta)^2 - 2\sqrt{R^2}\Phi^{-1}(\alpha)\Phi^{-1}(\beta) &\geqslant R^2\Phi^{-1}(\alpha)^2 + \Phi^{-1}(\beta)^2 - 2\sqrt{R^2}\Phi^{-1}(\alpha)\Phi^{-1}(\beta) \\
&= (\Phi^{-1}(\beta) - \sqrt{R^2}\Phi^{-1}(\alpha))^2 \geqslant 0
\end{aligned}
$$

Similarly,

$$
\Phi^{-1}(\alpha)^2 + \Phi^{-1}(\beta)^2 - 2\sqrt{R^2}\Phi^{-1}(\alpha)\Phi^{-1}(\beta) \geqslant (\sqrt{R^2}\Phi^{-1}(\beta) - \Phi^{-1}(\alpha))^2 \geqslant 0
$$

Thus,

$$
\begin{aligned}
\frac{\partial}{\partial R^2} V(\alpha, \beta, R^2) &\leqslant \frac{1}{4\pi\beta\sqrt{R^2(1-R^2)}} \exp\left(-\frac{1}{2(1-R^2)}(\Phi^{-1}(\beta) - \sqrt{R^2}\Phi^{-1}(\alpha))^2\right) \\
&= \frac{1}{\beta\sqrt{8\pi R^2(1-R^2)}} \phi\left(\frac{\Phi^{-1}(\beta) - \sqrt{R^2}\Phi^{-1}(\alpha)}{\sqrt{1-R^2}}\right)
\end{aligned}
$$

and

$$
\frac{\partial}{\partial R^2} V(\alpha, \beta, R^2) \leqslant \frac{1}{\beta\sqrt{8\pi R^2(1-R^2)}} \phi\left(\frac{\sqrt{R^2}\Phi^{-1}(\beta) - \Phi^{-1}(\alpha)}{\sqrt{1-R^2}}\right)
$$

∎

**Lemma D.4.** *The ratio*

$$\frac{\Phi\left(z\right)}{\phi\left(z\right)} \tag{8}$$

*is increasing in $z$.*

*Proof.* We compute the derivative of the ratio $\frac{\Phi(z)}{\phi(z)}$,

$$\frac{\partial}{\partial z} \frac{\Phi\left(z\right)}{\phi\left(z\right)} = \frac{\phi^2\left(z\right) + z\phi\left(z\right)\Phi\left(z\right)}{\phi^2\left(z\right)} = 1 + z\frac{\Phi\left(z\right)}{\phi\left(z\right)}$$

For $z \geqslant 0$ the derivative is clearly positive. For $z < 0$ we start by rewriting,

$$1 + z\frac{\Phi\left(z\right)}{\phi\left(z\right)} = \frac{z}{\phi\left(z\right)}\left(\frac{\phi\left(z\right)}{z} + \Phi\left(z\right)\right)$$

Since $\frac{\partial}{\partial z}\left(\frac{\phi(z)}{z} + \Phi\left(z\right)\right) = \frac{-z^2\phi(z)-\phi(z)}{z^2} + \phi\left(z\right) = \frac{-\phi(z)}{z^2} < 0$ and $\frac{\phi(z)}{z} + \Phi\left(z\right) \overset{z\to-\infty}{\to} 0$, it follows for $z < 0$ that $\left(\frac{\phi(z)}{z} + \Phi\left(z\right)\right) < 0$. Thus for any $z$,

$$\frac{\partial}{\partial z} \frac{\Phi\left(z\right)}{\phi\left(z\right)} \geqslant 0$$

∎

