# OpenReview forum: "The Value of Prediction in Identifying the Worst-Off"
_ICML.cc/2025/Conference — ICML 2025 oral_

### Official Review · Reviewer_FwzC · 2025-03-09

**Overall Recommendation:** 3

**Summary:**

The paper investigates the welfare impacts of using machine learning prediction systems in equity-driven government programs, particularly in identifying and supporting the most vulnerable individuals. The authors develop a framework to evaluate the relative effectiveness of prediction systems compared to other policy levers, such as expanding bureaucratic capacity.

**Claims And Evidence:**

The claims made in the paper are generally well-supported by both theoretical and empirical evidence. The authors provide clear mathematical formulations and proofs to support their theoretical claims, particularly regarding the prediction-access ratio (PAR) and the conditions under which prediction improvements are most effective. The empirical case study on long-term unemployment in Germany is well-designed and provides convincing evidence that expanding screening capacity often has a greater impact than improving prediction accuracy, especially in resource-constrained settings.

However, one potential limitation is that the empirical results are based on a single case study (long-term unemployment in Germany), which may limit the generalizability of the findings to other contexts. While the authors provide a theoretical framework that is broadly applicable, additional empirical validation across different domains would strengthen the paper's claims.

**Essential References Not Discussed:**

Relevant works that could be cited include:

Fairness in Machine Learning: The paper could discuss recent work on fairness-aware machine learning, such as Hardt et al., 2016, which provides a framework for evaluating and mitigating bias in predictive models.

Algorithmic Fairness in Public Policy: The authors could also cite Chouldechova et al., 2018, which discusses the trade-offs between accuracy and fairness in algorithmic decision-making, particularly in the context of child welfare screening.

**Experimental Designs Or Analyses:**

The experimental design and analysis are sound. The authors use a large, real-world dataset on German jobseekers to evaluate their framework, and they carefully control for potential confounding factors by focusing on unemployment spells that began between 2010 and 2015. The use of CatBoost for predicting unemployment durations is appropriate, and the authors provide a clear methodology for simulating improvements in prediction accuracy by scaling residuals.

One potential limitation is that the authors do not explore the impact of different model architectures or feature sets on the prediction accuracy. While they compare a shallow decision tree to CatBoost, a more comprehensive comparison of different models could provide additional insights into the trade-offs between model complexity and prediction accuracy.

**Methods And Evaluation Criteria:**

The proposed methods and evaluation criteria are appropriate for the problem at hand. The authors use a combination of theoretical modeling and empirical analysis to evaluate the effectiveness of prediction systems in identifying the worst-off. The prediction-access ratio (PAR) is a novel and useful metric for comparing the relative benefits of improving prediction accuracy versus expanding screening capacity. The empirical evaluation is based on a large, real-world dataset, and the authors use standard machine learning techniques (e.g., CatBoost) to predict unemployment durations, which is appropriate for the task.

One potential improvement could be the inclusion of additional evaluation metrics, such as fairness or equity considerations, given that the paper focuses on equity-driven programs. While the authors discuss the importance of identifying the worst-off, they do not explicitly evaluate whether their prediction systems disproportionately affect certain demographic groups.

**Other Comments Or Suggestions:**

N/A

**Other Strengths And Weaknesses:**

N/A

**Questions For Authors:**

The empirical results are based on a single case study. Could the authors comment on the generalizability of their findings to other domains, such as poverty targeting or social housing? How might the PAR differ in these contexts?

The paper focuses on identifying the worst-off, but does not explicitly address potential biases or disparities in the prediction systems. Have the authors considered the fairness implications of their framework, particularly in terms of how different demographic groups might be affected by the screening policies?

The authors compare a shallow decision tree to CatBoost, but do not explore other model architectures or feature sets. Could the authors provide additional insights into how different models might impact the prediction-access ratio (PAR) and policy value?

The paper discusses the relative value of improving prediction accuracy versus expanding screening capacity, but does not provide a detailed cost-benefit analysis. Could the authors provide more information on the costs associated with improving prediction accuracy (e.g., data collection, model training) versus expanding screening capacity?

**Relation To Broader Scientific Literature:**

The paper is well-situated within the broader scientific literature on machine learning for public policy and resource allocation. The authors draw on prior work in decision theory, operations research, and causal inference to develop their framework, and they cite relevant studies on the use of risk prediction systems in government programs. The paper extends prior work by focusing on the relative value of prediction in identifying the worst-off, rather than optimizing aggregate outcomes.

The authors also discuss recent work on the trade-offs and pitfalls of risk-scoring systems, which is an important consideration in the design of equity-driven programs. However, they could have more explicitly connected their work to the growing literature on fairness and bias in machine learning, particularly in the context of public policy.

**Theoretical Claims:**

The theoretical claims in the paper are well-supported by mathematical proofs and derivations. The authors provide detailed proofs for their key results, including the conditions under which the prediction-access ratio (PAR) is maximized and the impact of expanding screening capacity versus improving prediction accuracy. The proofs are sound and correctly derived, and the authors provide intuitive explanations for their results.

One minor issue is that some of the theoretical results are presented informally (e.g., Theorem 1.1 and Theorem 1.2), which could make it difficult for readers to fully understand the technical details.

---

> ### Author Rebuttal · Authors · 2025-03-31
>
> Thank you for taking the time to carefully read our manuscript and provide comments. It is very much appreciated.
>
> Other Domains. Yes, performing similar evaluations in other empirical domains would be valuable, but beyond the scope of this paper due to the substantial effort required to access administrative, government datasets. While each context should be analyzed individually, the close alignment between our theoretical results and empirical findings might suggest that the PAR could behave similarly across different settings. Our paper exemplifies how to do this for a specific real-world, high-stakes setting and we hope it encourages future work in other domains.
>
> Fairness Evaluation. Thank you for raising this point. We fully agree that fairness concerns, such as differential performance across protected subgroups (e.g., gender), must be considered in the design of allocation systems. We view our analysis of policy levers as a necessary—but not sufficient—step toward understanding when investments in predictive systems are justified. We believe these subgroup concerns are an important direction for follow up work.
>
> We appreciate your suggestion of relevant references and will more explicitly connect our work to the literature on fairness in algorithmic decision-making (including Hardt et al., 2016 and Chouldechova et al., 2018) in the revised related work section. We will also include additional figures in the revision that show model performance across demographic subgroups.
>
> Additional Models. We selected CatBoost as it is widely regarded as a strong, out-of-the-box model for tabular data. Government agencies often rely on standard, readily available ML models, and thus our primary aim was not to optimize predictive accuracy but rather to illustrate trade-offs that arise with commonly used predictive methods. Our findings remain robust when using alternative models, such as random forests.
>
> Costs. Thanks for bringing this up. The costs of improving prediction accuracy versus expanding screening capacity will typically differ and be domain-specific. A complete cost-benefit analysis would multiply the PAR by the ratio of marginal costs of both policy levers. Policymakers often have clearer estimates of these direct costs, while the PAR helps them determine how much more they should be willing to pay for one lever relative to the other. We will expand these cost considerations in the introduction of our revised manuscript.
>
> Presentation. We will streamline the introduction and link the informally stated main results to their formal counterparts to ensure technical clarity.

---

### Official Review · Reviewer_aH4L · 2025-03-10

**Overall Recommendation:** 4

**Summary:**

In this paper, the authors consider a scenario where, for example, a government agency wants to identify the x% worst off in a population, based on a predicted metric $Y$. Once we have the predicted $Y$ for the population, we can sort everyone and have a follow-up check on those that scored worst. However, there is a budget $\alpha$ for how many follow-up checks we can do. The authors argue that we should mostly care about the true positive rate in this case: how many of the people in the x% worst off have we sent to the follow-up check? We have thus two levers to affect the result: we can improve the predictions for $Y$ or we can have more follow-up checks. The paper asks the question when you should do which and makes some recommendations.

**Claims And Evidence:**

The core of the paper is a mathematical model of how to determine whether there is more value in improving predictions or more value in increasing the follow-up testing budget.

I think the framework they define makes sense and is mostly well justified. This is mostly quite a good paper.

There is just one place where I think the claims are at the very least misleading; maybe even wrong: the fact that $\Delta_{R^2}$ and $\Delta_\alpha$ are treated as having the same "unit". There is a paragraph on page 5, column 2, lines 251-264, which admits that this is wrong to do; it admits that one should consider the costs of $\Delta_{R^2}$ and $\Delta_\alpha$ instead, but then the rest of the paper treats $\Delta_{R^2}$ and $\Delta_\alpha$ as comparable anyway – for example, in Figure 3, you draw a line at PAR=1 as if that were a meaningful threshold in any way!

I would think PAR should be defined like this:

$$
\mathrm{PAR} = \frac{\frac{\partial V}{\partial\alpha}\frac{\mathrm{d}\alpha}{\mathrm{d} c}}{\frac{\partial V}{\partial R^2}\frac{\mathrm{d}R^2}{\mathrm{d} c}}
$$

Where $c$ is the total cost. This definition is robust to change of variables. For example if you were to go from $R^2$ to $2R^2$.

I understand why the authors didn't want to do it like this – it means having to model the total cost as well – but right now it just seems wrong to me.

---

I would also have wished for more of a direct contrasting with Perdomo (2024). Why make different choices than that paper?

**Essential References Not Discussed:**

None that I'm aware of.

**Experimental Designs Or Analyses:**

The numerical simulations and the case study are very well done. Here the paper really shines.

**Methods And Evaluation Criteria:**

There isn't really a *method* proposed in the paper, so I can't really judge that. One thing that is maybe a bit odd is that when $R^2$ (the coefficient of determination) is picked as the variable that tracks how good the prediction model is, there is no discussion of any alternative variables one could choose and $R^2$ is then used even in the theorems.

**Other Comments Or Suggestions:**

- I would like it if the figures with multiple plots had subcaptions so that they can be better read at a glance.

**Other Strengths And Weaknesses:**

## Weaknesses in the presentation

- To me, section 1 is too long and too repetitive. The authors also made the choice of first explaining many things informally in section 1 and then explaining them formally in section 2. This, to me, was very confusing and I would have preferred reading the formal explanations directly. Reading the informal explanations first did not help me.
  * For example, on page 2, PAR is defined informally, but all I kept wondering was how you want to compute it, which is then finally revealed on page 4. Maybe at least reference the later definition in the first informal definition.
  * Informal Theorem 1.1 uses $R^2$ without any previous mention of it. Just please at least briefly define it before the theorem (or better yet, scrap the whole informal theorem stuff)
  * Even when $R^2$ is finally defined on page 4, it happens *after* Proposition 2.3, which also references it
- There is no Discussion/Conclusion at the end of the paper. I think even just a paragraph would be nice here. You can easily get enough space for this by trimming section 1.

**Questions For Authors:**

1. Is there a justification for why PAR=1 is treated as a significant threshold even though it seems to me to be a pretty arbitrary one?
2. You write “rather than evaluating changes in overall expected welfare \[like Perdomo (2024) does\], we measure the fraction of truly worst-off individuals who are identified”; can you expand on why you deviate from Perdomo (2024) here?

**Relation To Broader Scientific Literature:**

The paper seems to build on Perdomo (2024) and seems to be a reasonable extension of that work, but I'm not really an expert in that field.

**Theoretical Claims:**

I did not check the correctness of the proofs, but the theorems all look very reasonable.

---

> ### Author Rebuttal · Authors · 2025-03-31
>
> Thank you for your detailed, thoughtful questions and suggestions for improving the paper! We appreciate your encouraging feedback on our conceptual framework and empirical analysis.
>
> Cost Ratio and PAR = 1. Thank you for raising this. We fully agree that $\Delta_{\alpha}$ and $\Delta_{R^2}$ represent changes with different costs.  For clarity and generality, we deliberately chose not to explicitly model these costs, as they naturally vary across application domains. We believe that examining the PAR independently remains valuable to disentangle benefits from costs. Policymakers typically find it easier to estimate relative costs of practical policy levers (e.g., hiring additional staff or conducting more frequent surveys to improve predictive accuracy) compared to evaluating how these changes will impact the policy value. We therefore focused on shedding insight on the welfare ratio with the understanding that domain experts have knowledge of the cost ratio and can easily factor these into the cost-benefit analysis (as in your expression).
>
> While our results hold for general $\Delta_\alpha$ and $\Delta_{R^2}$, we chose PAR =1 with these fixed ($\Delta_\alpha = \Delta_{R^2}$) purely for the sake of discussion. We are happy to point out how this choice is made for simplicity and will add further discussion on how the true threshold is at (cost of prediction) / (cost of access) as you point out. In the final version, we will move up and expand the discussion on marginal costs to the introduction, and we will aim to provide further context on the PAR = 1 threshold to avoid any confusion. Your perspective and comments on this issue are very helpful. Thank you.
>
> Why differ from Perdomo 2024. This previous work focuses on the impact of different policy levers on *average* social welfare. By contrast, we focus on identifying the impacts of various policy levers on the welfare of the *worst-off* individuals. This captures a (mathematically and conceptually) distinct as well as a practically important setting often seen in the public sector. For instance, employment agencies commonly aim to identify and support those in greatest need, rather than maximizing average employment outcomes across all job seekers. We will clarify this distinction from Perdomo (2024) in the related work section of the revised manuscript.
>
> Choice of $R^2$. We will clarify the choice of $R^2$ and discuss other measures that could also be applicable in the final version of the manuscript. The short answer is that $R^2$ tends to exactly characterize predictability in our theoretical models.
>
> Presentation. Thank you for your suggestions on improving the presentation—your comments are very helpful in this regard. We will streamline the introduction to reduce redundancy and clarify the presentation to ensure that key concepts are formally defined before they are used. Given the additional space that comes with the revision, we will happily include a discussion at the end of the paper and will include subcaptions for the figures in the revised version of the manuscript.

---

### Official Review · Reviewer_gDhp · 2025-03-13

**Overall Recommendation:** 4

**Summary:**

This paper investigates the value of ML prediction when the goal is to identify the worst candidates as a screening problem. Formulating the problem as a screening problem with threshold, this paper derives 2 main theorems that characterizes the behavior of screening policies in general, and complement the studies with empirical evaluation whose outcome largely matches the theoretical derivation.

**Claims And Evidence:**

Yes

**Essential References Not Discussed:**

N/A

**Experimental Designs Or Analyses:**

Yes, the design is consistent with the theoretical setup.

**Methods And Evaluation Criteria:**

Yes

**Other Comments Or Suggestions:**

N/A

**Other Strengths And Weaknesses:**

Strengths: the paper is well-written, with the motivation of the problem as well as intuition of solution clearly explained. The problem investigated is of high relevance in real-time application of machine learning to social problems.

**Questions For Authors:**

N/A

**Relation To Broader Scientific Literature:**

This paper investigates the value of ML-based policies in the context of screening.

**Theoretical Claims:**

N/A. Not familiar with field - intuitively sensible.

---

> ### Author Rebuttal · Authors · 2025-03-31
>
> Thank you very much for your positive feedback. We're especially glad you found the paper clear and relevant, and appreciate the time you took to engage with our work.

---

### Official Review · Reviewer_V6gy · 2025-03-13

**Overall Recommendation:** 4

**Summary:**

The authors study the question when and whether policymakers should invest into improving their predictions or expanding screening, with respect to bottom-line welfare. They set up a simple theoretical model, and validate the resulting hypothesis in semi-synthetic experiments on the German labor market.

**Claims And Evidence:**

They claim to identify when improving predictions provides a higher benefit in helping identify the worst-off individuals, relative to expanding screening of the population, namely when the prediction system explains either none or almost all of the variance in the outcome (measures in $R^2$). They support this claim via theoretical findings on a simplistic model (Gaussian outcome & residuals), and in experiments on real-world data.

**Essential References Not Discussed:**

No.

**Experimental Designs Or Analyses:**

The experimental setup is clear. However, the figures always show the results for different values of $\beta$, although it was argued that $\beta$ is 0.15 at the beginning. This confuses me and made them hard to read, especially Figure 4.

**Methods And Evaluation Criteria:**

They define welfare/value of a prediction system as the fraction of the at-risk population identified. This seems like a plausible definition for applications that focus on the worst-off individuals.

**Other Comments Or Suggestions:**

The considerations for incremental improvements of the prediction model in Section 4 seems out of context. The authors say that it allows them to gauge how similar but slightly better models affect policy outcomes, but the actual follow-up is only in the empirical section.

**Other Strengths And Weaknesses:**

The paper is well-written, and addresses an important question for policymakers.

**Questions For Authors:**

1. Could you please differentiate which ideas are truly new and which are adapted from Perdomo, 2024?
2. Could you please explain Figure 4(c)?

**Relation To Broader Scientific Literature:**

The work extend existing considerations from Perdomo, 2024. The main difference seems to be the definition of the welfare, here fraction of the at-risk population vs linear/probit model. It would be helpful if the authors are more clear with ideas are truly their and which are adapted to their setting.

Perdomo, J. C. (2023). The relative value of prediction in algorithmic decision making. arXiv preprint arXiv:2312.08511.

**Theoretical Claims:**

I checked the correctness of Proposition 2.2. and 2.3. I merely skimmed over the others, it seemed plausible though.

---

> ### Author Rebuttal · Authors · 2025-03-31
>
> Thank you for engaging with our work and for your constructive feedback. We very much appreciate it!
>
> Beta Values. 0.15 reflects the official government threshold used in Germany. However, we included additional values to account for differing long-term unemployment thresholds used in other countries. To improve clarity, we will reorganize the figures and expand the captions in the revision.
>
> Comparison to Perdomo 2024. Thank you for the question. Our work conceptually builds on Perdomo (2024), who introduced the prediction-access ratio (PAR) to analyze trade-offs between prediction quality and resource availability. The main difference between our work and his is that we develop a new theoretical analysis of the PAR focused specifically on the welfare of the *worst-off* population rather than *average* welfare, as in Perdomo 2024. These two settings are conceptually and mathematically distinct.  Furthermore, relative to Perdomo 2024, we introduce a set of empirical tools to analyze these tradeoffs in practice (the previous work was purely theoretical). We will clarify and expand the comparison to Perdomo (2024) in the revised related work section.
>
> Section 4. We are happy to improve the transition into Section 4 and connect it more explicitly to the empirical case study in the revised manuscript.
>
> Figure 4(c). Figure 4(c) shows the rate of improvement in policy value with respect to small changes in prediction quality in a high $R^2$ regime (predictions close to perfect), as a function of available resources $\alpha$. Incremental prediction improvements have the greatest impact when resources precisely match the targeted fraction of the population ($\alpha = \beta$). For both insufficient ($\alpha < \beta$) and surplus ($\beta > \alpha$) resources, further prediction improvements are less impactful. We will expand and improve the caption for Figure 4 in the final version. Thanks for raising this.

---

### Decision · Program_Chairs · 2025-05-01

**Decision:**

Accept (oral)

**Comment:**

This paper presents a stylized model of policy problems where the goal is to identify applicants to a program who are the worst-off. It studies the marginal value of investing resources in improving a predictive model for this task compared to expanding the capacity to screen more applicants. The focus of the paper is on both theoretically characterizing this tradeoff as well as an empirical case study of an unemployment program. Reviewers found the paper to be an important contribution to the emerging literature on the value and tradeoffs associated with the use of prediction and social settings. I also believe it is important for the field to pay more attention to these issues.

I do agree with the suggestion made by Reviewer aH4L, that the "units" of changes in prediction vs access are not the same. While the authors are correct that costs are both application-specific and typically easier to determine, the right interpretation of the tradeoff might vary substantially depending on both much we expect them to differ (eg if in some cases improving prediction can be viewed as a fixed cost, vs a marginal cost for each additional applicant screened). I encourage the authors to expand their discussion of this point in the revised paper.